# Length-dependent disassembly maintains four different flagellar lengths in *Giardia*

Shane G McInally[1], Jane Kondev[2], Scott C Dawson[1]*

[1]Department of Microbiology and Molecular Genetics, University of California, Davis, Davis, United States; [2]Department of Physics, Brandeis University, Waltham, United States

**Abstract** With eight flagella of four different lengths, the parasitic protist *Giardia* is an ideal model to evaluate flagellar assembly and length regulation. To determine how four different flagellar lengths are maintained, we used live-cell quantitative imaging and mathematical modeling of conserved components of intraflagellar transport (IFT)-mediated assembly and kinesin-13-mediated disassembly in different flagellar pairs. Each axoneme has a long cytoplasmic region extending from the basal body, and transitions to a canonical membrane-bound flagellum at the 'flagellar pore'. We determined that each flagellar pore is the site of IFT accumulation and injection, defining a diffusion barrier functionally analogous to the transition zone. IFT-mediated assembly is length-independent, as train size, speed, and injection frequencies are similar for all flagella. We demonstrate that kinesin-13 localization to the flagellar tips is inversely correlated to flagellar length. Therefore, we propose a model where a length-dependent disassembly mechanism controls multiple flagellar lengths within the same cell.

**\*For correspondence:**
scdawson@ucdavis.edu

**Competing interests:** The authors declare that no competing interests exist.

## Introduction

Eukaryotic flagella and cilia (used interchangeably) are dynamic, compartmentalized microtubule (MT) organelles that facilitate motility and chemosensation, and direct hydrodynamic flow during development (*Brooks and Wallingford, 2014*; *Pazour and Witman, 2003*). Over 500 distinct proteins comprise the highly conserved axoneme architecture, defined by nine MT doublets surrounding a central MT pair ('9+2') (*Ishikawa, 2017*). Axonemes are nucleated at basal bodies and extend from the complex transition zone (TZ), which acts as a diffusion barrier at the base of the membrane-bound cilium (*Reiter et al., 2012*). Assembly and maintenance of flagellar length is dependent upon bidirectional MT motor-driven intraflagellar transport (IFT) to provide building blocks to the site of assembly, the distal flagellar tip (*Kozminski et al., 1993*; *Marshall and Rosenbaum, 1999*). Canonical axoneme architecture, as well as IFT components and flagellar assembly mechanisms, likely predate the radiation of all extant lineages (*Ishikawa, 2017*; *Sung and Leroux, 2013*) due to widespread conservation in diverse unicellular flagellates ranging from *Chlamydomonas, Tetrahymena*, and *Trypanosoma* to the ciliated cell types of invertebrates and mammals (*Buisson et al., 2013*; *Hao and Scholey, 2009*; *Kozminski et al., 1993*).

Yet despite these conserved elements, there is considerable variation in flagellar number, structure and function in microbial and multicellular eukaryotes (*Avidor-Reiss et al., 2017*; *Ishikawa, 2017*). Many eukaryotes possess from two to thousands of cilia with unique functions, lengths, morphologies, or inheritance patterns (e.g., the multiciliated protozoan *Tetrahymena* or multiciliated human epithelial cells). Differences in IFT-dependent ciliogenesis mechanisms (*Brooks and Wallingford, 2014*; *Ishikawa, 2017*) can lead to hallmark variations in axoneme number and structure in many metazoan cell types, as well as the atypical axoneme structures found in many flagellated protists. Conversely, the canonical '9+2' axoneme structure in metazoan sperm or

apicomplexan parasites can be assembled through alternative IFT-independent 'cytosolic ciliogenesis' mechanisms (*Avidor-Reiss and Leroux, 2015*).

How does such flagellar structural and functional variation arise from conserved IFT components and assembly/maintenance mechanisms? The parasitic protist *Giardia lamblia* is an ideal model to investigate how unique flagellar types and flagellar lengths are built and maintained within a single multiciliated cell. *Giardia* has eight flagella organized as four bilaterally symmetric pairs. The eight flagella have four different equilibrium lengths, which implies a regulatory mechanism to sense and differentially modulate assembly or disassembly rates between the different flagellar pairs. Equilibrium axoneme lengths of all eight flagella are also sensitive to MT stabilizing or depolymerizing drugs (*Dawson et al., 2007*). While each of the eight axonemes retains the characteristic '9+2' MT architecture, each axoneme also has a cytoplasmic, non-membrane-bound region that extends from a centrally located basal body before exiting the cell body as a membrane-bound flagellum (*McInally and Dawson, 2016*). *Giardia* flagella lack a transition zone or TZ protein homologs, yet the genome encodes homologs of IFT train components (IFT-A and IFT-B proteins), kinesin-2 and IFT dynein motors (*Avidor-Reiss and Leroux, 2015*; *Barker et al., 2014*). Similar to other eukaryotes, kinesin-2 and kinesin-13 regulate flagellar assembly and disassembly in *Giardia*, as CRISPRi-based knockdowns or the overexpression of dominant negatives result in dramatic flagellar length defects (*Dawson et al., 2007*; *McInally et al., 2019*). Lastly, during cell division, four intact mature axonemes and basal bodies are structurally inherited (anterior and caudal) and four new axonemes are assembled de novo in each daughter cell (*Hardin et al., 2017*; *Nohynková et al., 2006*). The cytoplasmic regions of the de novo posteriolateral and ventral axonemes are assembled prior to cytokinesis (*Hardin et al., 2017*).

Flagella are informative models to study organelle size control, as each flagellum maintains a consistent equilibrium length with a single dimension to represent its size (*Marshall et al., 2005*; *Tamm, 1967*). The classic 'long-zero' experiment in *Chlamydomonas* demonstrated length equalization of both flagella when one was amputated, and thus implies that a limited precursor pool is shared between flagella (*Coyne and Rosenbaum, 1970*). Built on this and other prior work, the prevailing explanation for the regulation of equilibrium flagellar length is the 'balance-point model', which argues that constitutively controlled steady-state length is a balance between a length-dependent assembly rate and a length-independent disassembly rate (*Marshall et al., 2005*). However, equilibrium length can be altered by modulating either the rates of flagellar assembly or disassembly (*Mohapatra et al., 2016*).

While IFT-mediated assembly has been presumed to be the primary driver of ciliary length (*Engel et al., 2009*; *Hendel et al., 2018*; *Ludington et al., 2015*), flagellar disassembly can also modify flagellar equilibrium length as defined by the balance point model. Kinesin-13 (and kinesin-8) were first identified as depolymerizers of cytoplasmic and spindle MTs, but later were discovered to depolymerize MTs at distal ciliary tips (*Helenius et al., 2006*; *Walczak et al., 2013*). In *Giardia* (*Dawson et al., 2007*) and in other microbial flagellates like *Leishmania*, kinesin-13 activity at the ciliary tips directly contributes to flagellar disassembly, yet may also indirectly impact IFT-mediated assembly by modulating cytoplasmic tubulin pools (*Blaineau et al., 2007*; *Chan and Ersfeld, 2010*; *Piao et al., 2009*; *Vasudevan et al., 2015*; *Wang et al., 2013*). In *Chlamydomonas*, kinesin-13 promotes disassembly of axonemes via IFT transport during the induction of flagellar resorption. Depletion of kinesin-13 prevents depolymerization of cortical MTs resulting in shorter flagella due to the reduction of the cytoplasmic tubulin pool required for IFT-mediated assembly (*Piao et al., 2009*; *Wang et al., 2013*).

How does the balance between IFT-mediated assembly and kinesin-13 mediated disassembly maintain four pairs of flagella with *different* equilibrium cytoplasmic and membrane-bound flagellar lengths in *Giardia*? Here we quantified the dynamics of IFT-mediated assembly in cytoplasmic and membrane-bound regions of flagella of different lengths. By tracking IFT particle behavior and turnover in live cells in unprecedented detail, we discovered that the eight flagellar pore regions act as the diffusion barriers for each flagellar compartment. Rather than at basal body or transition zone regions, IFT proteins diffuse bidirectionally on the cytoplasmic regions of the axonemes and accumulate at each flagellar pore, where IFT trains are presumably assembled and injected into the membrane-bound axonemes. IFT train speed, size, and frequency of injection are similar regardless of the length of the flagellar pair, and increasing flagellar lengths with the MT-stabilizing drug Taxol did not change the injection rate of IFT. We further show that kinesin-13 accumulates in a length-

dependent manner to the flagellar tips. Lastly, we propose a model for *Giardia* flagellar length regulation that emphasizes the length-dependent disassembly process to balance a length-independent IFT injection rate for each flagellar pair. The unique architecture and varied equilibrium lengths of *Giardia's* eight flagella challenge the canonical models of IFT-mediated flagellar assembly and length regulation. We anticipate that other flagellated microbes or multicellular cell types may regulate flagellar lengths with similar length-dependent disassembly mechanisms.

## Results

### Four different equilibrium lengths of cytoplasmic and membrane-bound axonemes

The four flagellar pairs in *Giardia* have distinct lengths for both cytoplasmic and membrane-bound regions of the axonemes. The cytoplasmic axonemal regions span from the basal body to flagellar pore (*Figure 1A*, shaded; *Figure 1B*) and membrane-bound axonemal regions span from the flagellar pore to flagellar tip (*Figure 1A*, colored). To quantify the average lengths of membrane-bound and cytoplasmic axoneme regions, we imaged fixed trophozoites expressing a single, integrated copy of mNeonGreen-tagged β-tubulin to mark the MT cytoskeleton (*Figure 1C*). Total axoneme lengths (measured from basal body to flagellar tip) vary between the four pairs. The anterior axonemes had an average length of 19.1 ± 0.4 µm, the caudal flagella were 20.5 ± 0.6 µm, the posteriolateral flagella were 16.2 ± 0.4 µm, and the ventral flagella were 16.8 ± 0.9 µm (*Figure 1—figure supplement 1*). We also confirmed the average lengths of membrane-bound regions in wild-type WBC6 trophozoites: anterior flagella were 12.8 ± 0.1 µm, caudal flagella were 8.1 ± 0.1 µm, posteriolateral flagella were 8.7 ± 0.2 µm, and ventral flagella were 13.7 ± 0.9 µm (*Figure 1D* and *Hoeng et al., 2008*).

### IFT homologs localize to both cytoplasmic and membrane-bound regions of all flagella

In *Giardia*, the basal bodies are physically separated from the membrane-bound regions of flagella by the cytoplasmic axoneme regions, a distance that varies from 3 to 12 µm between the various flagellar pairs (*Figure 1—figure supplement 1*). Due to the unique architecture of *Giardia's* flagella, we wanted to first define the location of IFT train assembly and injection. Previously, components of IFT-A (IFT140), IFT-B (IFT81), and kinesin-2 were localized to both the cytoplasmic and membrane-bound axonemal regions (*Hoeng et al., 2008*). To confirm and extend this prior work, we imaged fluorescently tagged C-terminal fusions of homologs of IFT (12), BBSome (3), and kinesin-2 proteins (2). All *Giardia* IFT-A (IFT121, IFT122, andIFT140) and IFT-B (IFT38, IFT54, IFT56, IFT57, IFT74/72, IFT80, IFT81, IFT88, and IFT172) homologs localized to the basal body region, flagellar pores, flagellar tips, and along the lengths of both the cytoplasmic and membrane-bound regions of all axonemes (*Figure 2A* and *Hoeng et al., 2008*). Anterograde kinesin-2a and kinesin-2b motors localized to the eight flagellar pores and flagellar tips, with less fluorescent signal on the cytoplasmic axonemes or basal bodies than tagged IFT proteins (*Figure 2A* and *Hoeng et al., 2008*). Of the three BBSome homologs in *Giardia*, BBS4 localized primarily to the flagellar pores, BBS5 localized primarily to the cytoplasmic axonemes, and BBS2 localized primarily to the cytoplasm with some localization to the cytoplasmic axonemes (*Figure 2A*). All IFT proteins densely localized to cytoplasmic regions of all eight axonemes and distinct puncta of IFT trains were not observed on any of the cytoplasmic axonemes.

To determine whether overexpression of IFT proteins from episomal plasmids prevents observation of discrete IFT trains on the cytoplasmic axonemes, we also integrated C-terminal mNeonGreen (mNG) and GFP tags into the native *Giardia* IFT81 locus (*Figure 2—figure supplement 1*). Strains expressing single, integrated copies of IFT81GFP or IFT81mNG had the same subcellular localization pattern as strains expressing tagged IFT81 from the episomal vector, but the labeling intensity was more uniform in the population of integrated transformants. IFT81mNG was at least 3-fold brighter than IFT81GFP (*Figure 2—figure supplement 2*).

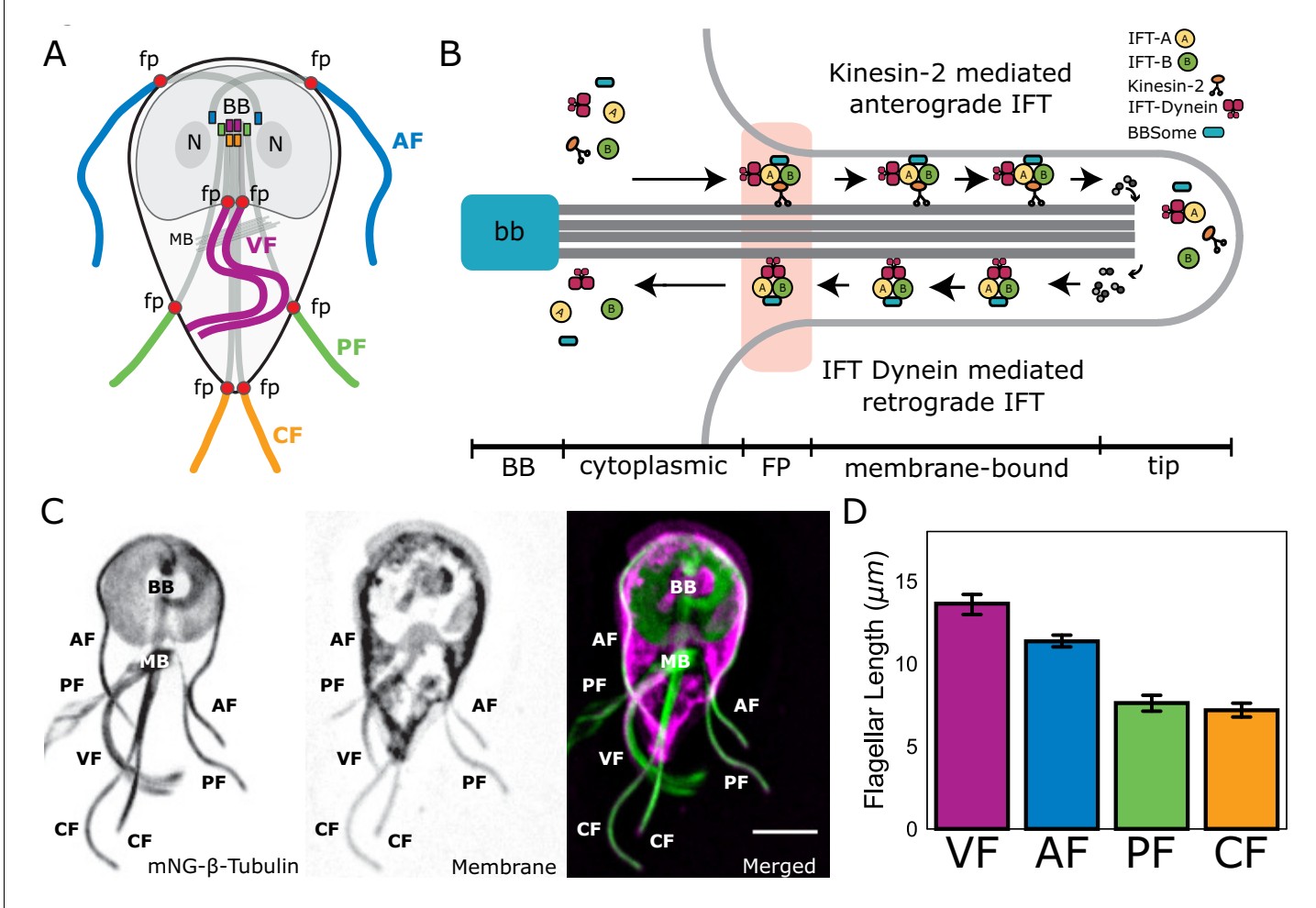

**Figure 1.** *Giardia* maintains four flagellar pairs with unique equilibrium lengths. (A) Schematic representation of membrane-bound, cytoplasmic, basal body (BB), and flagellar pore (fp) regions of the axoneme, as well as the two nuclei (N) and median body (MB). (B) Schematic representation of the specific regions of *Giardia*'s flagellar axoneme, including the basal body (BB), cytoplasmic axoneme (cytoplasmic), flagellar pore (FP), membrane-bound axoneme (membrane-bound), and the flagellar tip (tip). (C) Fluorescent labeling of the microtubule cytoskeleton and membrane of a *Giardia lamblia* trophozoite, including the median body (MB), the basal body (BB), and the four flagellar pairs: anterior (AF), posteriolateral (PF), caudal (CF), and ventral (VF). Scale bar, 5 μm. (D) Flagellar length quantification of membrane-bound regions of flagellar pairs of *Giardia* WBC6 trophozoites. The 95% confidence interval and average length are indicated. n ≥ 35 flagella for each pair. All pairs are statistically significantly different (p≤0.05, t-test) in membrane-bound length, except the posteriolateral and caudal flagella.

The online version of this article includes the following figure supplement(s) for figure 1:

**Figure supplement 1.** Quantification of full axoneme lengths in *Giardia lamblia*.

## IFT proteins accumulate at the flagellar pore regions of all flagella

To determine the spatial distribution of IFT proteins along the entire length of the flagellum, we used line scans to trace IFT81mNG fluorescence from the basal body to the flagellar tip (*Figure 2B and C*). Our analyses of cytoplasmic axonemes were limited to the anterior and posteriolateral flagella, as we were unable to reliably measure fluorescence intensity from the cytoplasmic regions of caudal and ventral flagella. IFT81mNG fluorescence was not uniformly distributed along the lengths of either the anterior or posteriolateral flagella (*Figure 2B*). The maximum fluorescence intensity for both flagella occurred at the flagellar pore, a region that lies at the transition from the cytoplasm to the compartmentalized flagellum (*Figure 2C*). IFT81mNG fluorescence in the anterior flagella had a single distinct maximum at the flagellar pore, whereas the posteriolateral flagella had one maximum at the posteriolateral flagellar pore and another maximum at a region adjacent to the ventral flagellar pores (*Figure 2C*). The cytoplasmic axoneme regions of both the anterior and posteriolateral

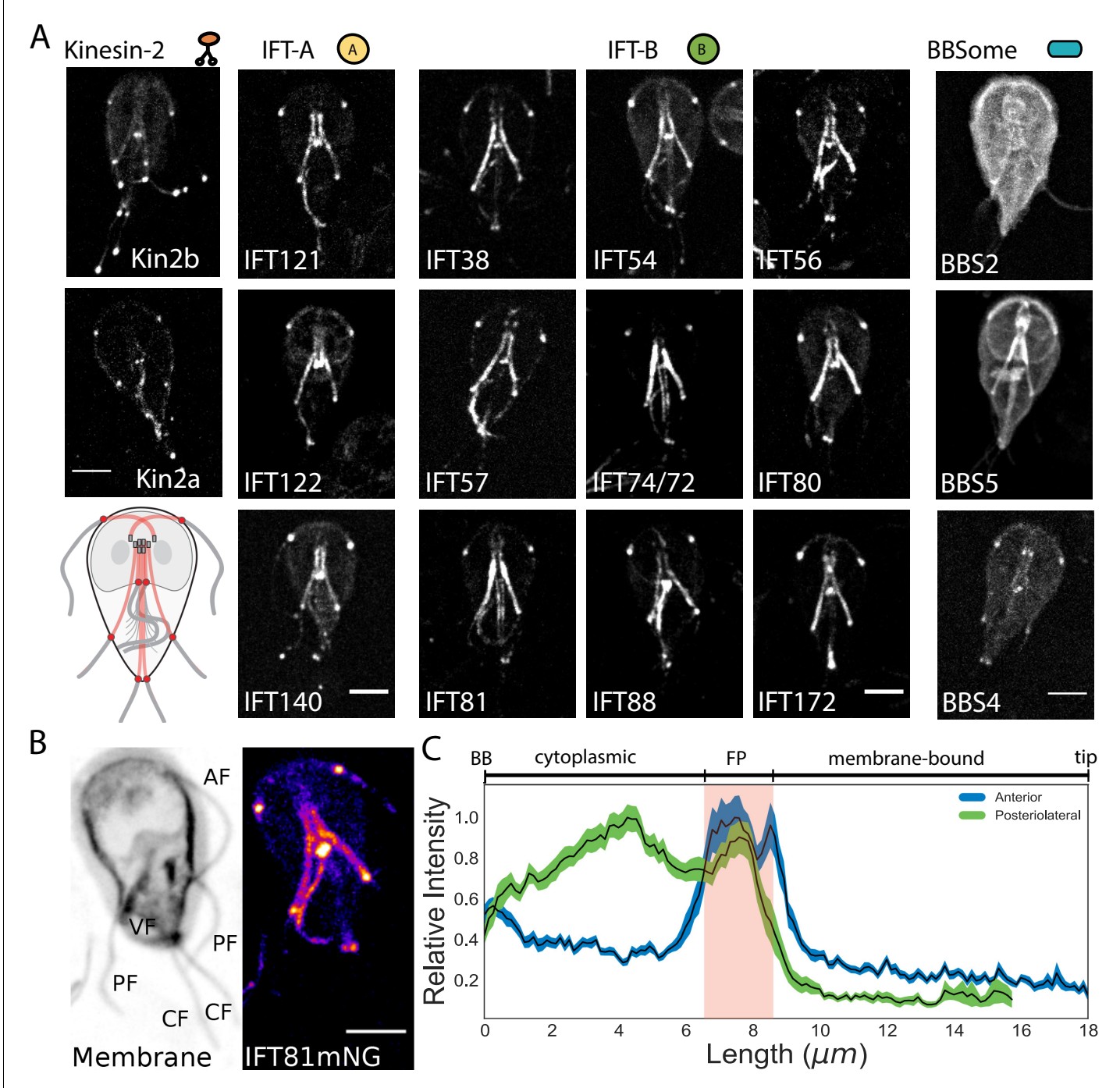

**Figure 2.** IFT proteins accumulate in the flagellar pore regions. (**A**) Maximum intensity projections of live cells show the distribution of kinesin-2, IFT-A complex, IFT-B complex, and BBSome proteins throughout the trophozoite. Representative schematic of IFT-A and IFT-B localizations is in the lowest left corner. All scale bars, 5 µm. (**B**) IFT81mNeonGreen proteins are more concentrated at the flagellar pore regions of the flagellar pairs. Scale bar, 5 µm. (**C**) Quantification of IFT81mNG distribution along the entire lengths of anterior and posteriolateral axonemes using line-scans. Black lines indicate mean intensity and shaded regions indicate 95% confidence intervals. Flagellar length is indicated on bottom axis and approximate anatomical position is indicated on the top axis, with red shading indicating the flagellar pore region. n = 31 for each flagellar pair, from four independent experiments. The online version of this article includes the following figure supplement(s) for figure 2:

**Figure supplement 1.** PCR validation of IFT81mNG and IFT81GFP integration into the native genomic locus.
**Figure supplement 2.** Comparison of brightness between IFT81GFP and IFT81mNG.

flagella had greater IFT81mNG fluorescence than the membrane-bound regions, but less fluorescence intensity than at flagellar pores. Furthermore, we did not observe differences between flagella of the same pair in these assays.

## Dynamic IFT protein localization at flagellar pore regions is driven by both diffusive and directed transport of IFT proteins

To understand how IFT proteins accumulate in the flagellar pore regions of flagella, we interrogated the behavior and turnover of IFT proteins associated with the flagellar pore regions using fluorescence recovery after photobleaching (FRAP). IFT proteins are dynamic at the flagellar pores, as indicated by FRAP of the posteriolateral flagella in the IFT81mNG strain (*Figure 3A*, *Video 1*). To determine the source of IFT proteins that facilitate this exchange, we photobleached IFT proteins that were associated with the posteriolateral cytoplasmic axonemes (*Figure 3B*, *Video 2*). IFT proteins were dynamic on cytoplasmic axonemes, and recovery is likely bidirectional as the photobleached region did not change position during recovery. To assess the contribution of IFT dynamics on the cytoplasmic axonemes to the accumulation of IFT proteins at the flagellar pore, we compared the rate of turnover for these two regions (*Figure 3C and D*). IFT turnover was approximately three times faster at the flagellar pore (effective diffusion constant = 0.049 ± 0.017 $\mu m^2 s^{-1}$) than in the cytoplasmic region (effective diffusion constant = 0.018 ± 0.005 $\mu m^2 s^{-1}$) (*Figure 3E*).

We hypothesized that the different rates of IFT protein turnover between the cytoplasmic and flagellar pore regions are due to two processes that promote the accumulation of IFT proteins at the flagellar pore: (1) diffusion of IFT proteins from the cytoplasmic axoneme region and (2) the return of IFT trains undergoing retrograde transport (*Figure 3F*). To test this hypothesis, we developed a model of IFT protein transport within the flagellar pore that incorporates both processes. Our model predicts a 3 ± 1 fold difference in the early (linear) speed of fluorescence recovery between the flagellar pore and cytoplasmic axoneme regions (Materials and methods). We measured the slope of the initial linear-phase of recovery for these two regions and found a 3.8 ± 0.7 fold difference (*Figure 3C and D*, dashed lines), consistent with our proposed model (*Figure 3F*).

## IFT particle size, frequency, and speed are similar between flagellar pairs of different lengths

Flagellar length control requires length-dependence of either the assembly rate, the disassembly rate, or both rates to establish an equilibrium length (*Mohapatra et al., 2016*). Thus, the maintenance of four different equilibrium flagellar lengths in *Giardia* could be due to different rates of IFT-mediated assembly or disassembly between flagellar pairs. To evaluate the contribution of IFT-mediated assembly to the maintenance of unique flagellar lengths, we quantified and compared IFT dynamics within the membrane-bound compartment of three of the four flagellar pairs. IFT dynamics were not possible to quantify on the ventral flagella, which continued to beat despite immobilization of trophozoites in low-melt agarose (*Figure 4A*, *Video 3*). Kymographs were used to compare intracellular anterograde (*Figure 4B*, magenta) and retrograde (*Figure 4B*, green) IFT dynamics between the anterior, posteriolateral, and caudal flagella (*Figure 4B*). For each flagellar pair, we quantified and compared the speed, size, and frequency of IFT trains undergoing transport within the membrane-bound flagellar compartment (*Figure 4C–F*) (*Engel et al., 2009*; *Mangeol et al., 2016*).

For all analyzed flagellar pairs there were no statistically significant differences between any of the measured IFT parameters. The average IFT train speed was consistent for both anterograde and retrograde directions, and IFT train speeds were not different between flagella of different lengths. Specifically, the average anterograde IFT train speed for all flagellar pairs was between 3.0–3.2 μm/sec, with no significant differences between any of the measured flagella (*Figure 4D*). Retrograde IFT velocity was not significantly different from anterograde velocity and the average retrograde IFT velocity (~3.2 μm/sec) was not significantly different between any of the measured flagellar pairs (*Figure 4E,F*).

The average size of anterograde IFT trains in anterior flagella was 24% larger than those in the shorter caudal flagella (*Figure 4C*), supporting a length-dependence of IFT train size for the anterior and caudal flagella. However, there were no significant differences between sizes of anterior and caudal retrograde IFT trains (*Figure 4E*). No significant differences in IFT train size were observed between anterior and posteriolateral flagella in either direction (*Figure 4D,F*).

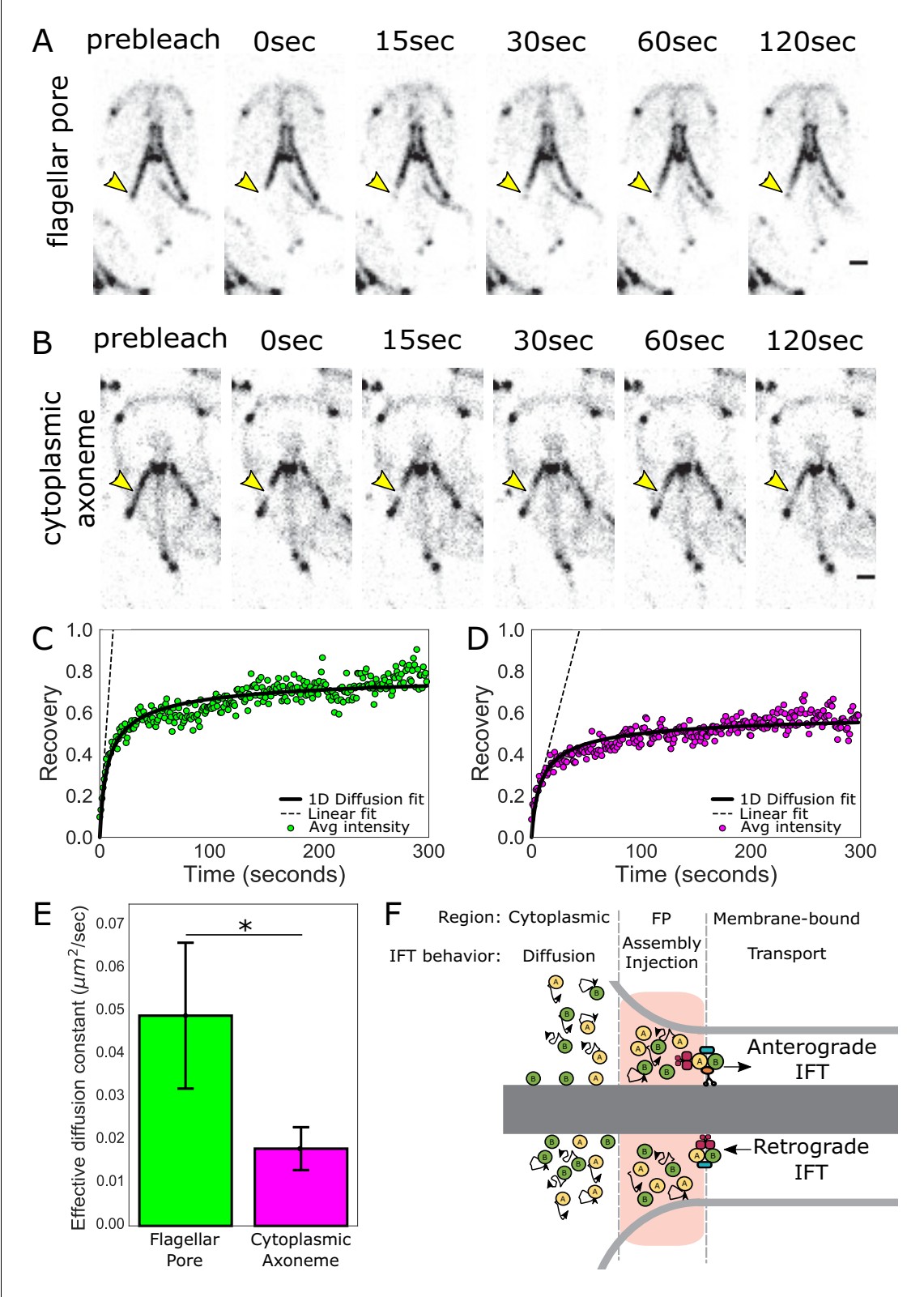

**Figure 3.** IFT train assembly occurs at the flagellar pore region. (**A**) Time series images of trophozoites expressing IFT81mNG prebleach, immediately post-bleach (0 s, yellow arrow) and during recovery (time in sec) for flagellar pore or (**B**) cytoplasmic axoneme regions. Scale bar, 2 μm. (**C**) Time averaged fluorescent recovery of posteriolateral flagellar pores and (**D**) cytoplasmic axonemes. Solid black lines indicate fit of the entire recovery phase. Dashed lines indicate linear fit of the initial recovery phase. n = 32 for bleached cytoplasmic axoneme regions, n = 25 for bleached flagellar pore

*Figure 3 continued on next page*

*Figure 3 continued*

regions, each from three independent experiments. (**E**) Effective diffusion constants from fitting FRAP recovery of the flagellar pore and cytoplasmic regions of posteriolateral flagella. Means and standard error are indicated. Student's t-test, *p=0.031. n ≥ 25 cells, from ≥three independent experiments. (**F**) Schematic representation of IFT particle behavior associated with the cytoplasmic axoneme, flagellar pore, and membrane-bound axoneme regions.

Comparisons of anterograde IFT injection frequency between anterior, caudal, and posteriolateral flagella showed that IFT train injection is not significantly different between flagella of different lengths (*Figure 4C and D*). Retrograde IFT frequency was also not significantly different between any of the flagella (*Figure 4E and F*). Using an alternative method to compare IFT injection frequency rates (trains/sec) between different flagella, we measured the time between each injection from kymographs filtered for only anterograde traffic (*Figure 4B*, magenta). The distribution of time-lag between injections is exponential for the three pairs analyzed, indicating a single rate limiting step for IFT train injection in the anterograde direction (*Figure 4G*). The frequency distribution was converted to a probability density function and a single exponential fit was used to measure the rate of injection (Materials and methods and *Figure 4G*). The average time between IFT train injections was similar between the three flagellar pairs: 1.0 ± 0.1 s for the anterior flagella; 1.3 ± 0.2 s for caudal flagella; and 1.2 ± 0.1 s for posteriolateral flagella (*Figure 4G*). Overall, the frequency of anterograde IFT train injection was not significantly different between longer (anterior) and shorter (caudal, posteriolateral) flagellar pairs.

## Perturbation of flagellar length supports the length-independence of IFT injection

To determine whether the total number of IFT trains within each flagellum scales linearly with length as predicted by a length-independent IFT injection rate, we used 3D structured illumination microscopy (3D-SIM) to quantify the total integrated intensity of IFT trains in fixed trophozoites expressing integrated IFT81GFP (*Figure 5A*). Fluorescence intensity of IFT81GFP and the length of the membrane-bound regions of flagella were measured using line scans. We observed a strong linear relationship between the total integrated fluorescence intensity and equilibrium flagellar length ($R^2$ = 0.89) (*Figure 5B*). Thus, the total amount of IFT trains within each *Giardia* flagellum scales linearly with length, supporting a constant IFT injection rate for each flagellar type.

*Giardia's* flagella are sensitive to treatment with the microtubule-stabilizing drug Taxol, which significantly increases the equilibrium flagellar length. Specifically, following one-hour incubation with 20 μM Taxol, the anterior flagella increased in length by 19%, the posteriolateral flagella by 27%, and the caudal flagella by 61% (*Figure 5C,D*). As Taxol treatment permits additional evaluation of the length-independence of IFT injection, the total integrated IFT intensity in elongated flagella was then quantified as previously described. Despite flagellar length increases, the direct relationship between the total amount of IFT trains in each flagellum and its length was unchanged after Taxol treatment (*Figure 5E*), which is consistent with the length-independent injection of IFT trains.

## Intensity of kinesin-13 localization at the flagellar tip is inversely correlated with flagellar length

In the absence of any significant differences in the assembly or maintenance of any of the flagella pairs analyzed, we investigated the possibility of length-dependent disassembly as a mechanism for maintaining four different equilibrium lengths. *Giardia* has a single kinesin-13

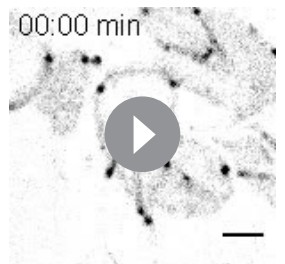

**Video 1.** Fluorescence recovery of IFT81mNG after photobleaching of anterior and posteriolateral flagellar pores. Fluorescence recovery following photobleaching of the right posteriolateral flagellar pore (top left) and right anterior flagellar pore (bottom right) in trophozoites expressing IFT81mNG. The video was recorded at one frame/second and is played at 10x increased speed. Time post-bleach (in minutes) is indicated in the top left corner. Scale bar, 5 μm.
https://elifesciences.org/articles/48694#video1

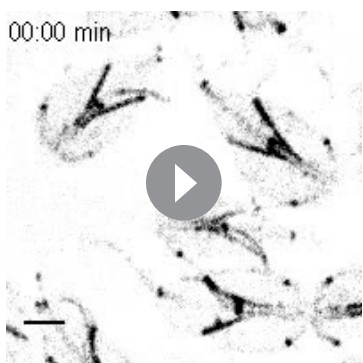

**Video 2.** Fluorescence recovery of IFT81mNG after photobleaching of posteriolateral cytoplasmic axonemes. Fluorescence recovery following photobleaching of the left posteriolateral cytoplasmic axoneme in trophozoites expressing IFT81mNG. The video was recorded at one frame/second and is played at 10x increased speed. Time post-bleach (in minutes) is indicated in the top left corner. Scale bar, 5 µm.
https://elifesciences.org/articles/48694#video2

homolog that localizes to all distal flagellar tips, the median body, and the two mitotic spindles (*Dawson et al., 2007*). We have shown previously that kinesin-13 regulates flagellar disassembly in *Giardia*, as the overexpression of kinesin-13 with a dominant negative rigor mutation and the depletion of kinesin-13 by CRISPRi-mediated knockdown caused increased caudal flagellar length (*Dawson et al., 2007*; *McInally et al., 2019*).

To assess the contributions of kinesin-13 mediated disassembly to the maintenance of flagellar length, we quantified the length of all four flagellar pairs in a CRISPRi-mediated kinesin-13 knockdown (K13kd) strain (*Figure 6A and B*). Here we extend prior work that characterized flagellar length defects in this strain (*McInally et al., 2019*) to show that all flagellar pairs have steady-state length increases when kinesin-13 expression is inhibited by ~60% (*Figure 6A and B*). Compared to non-specific gRNA controls, the length of anterior flagella in the K13kd strain is increased by an average of 1.1 ± 0.04 µm, the posteriolateral by 0.9 ± 0.03 µm, the caudal by 3.1 ± 0.18 µm, and the ventral by 1.6 ± 0.06 µm (*Figure 6B*).

To characterize the localization of kinesin-13 in live trophozoites, we constructed a strain expressing kinesin-13 with a C-terminal mNeonGreen tag (kinesin-13mNG). Kinesin-13mNG localized to all interphase microtubule structures, including the median body, the ventral disc, and distinct regions of all eight flagella. In contrast to the uniform kinesin-13mNG localization on the disc or median body, kinesin-13mNG was distributed unevenly along all flagella, localizing primarily to the distal flagellar tips, but also localizing to cytoplasmic axonemes and flagellar pores (*Figure 6C*). Using line-scans from the flagellar tips to the flagellar pores, we measured the spatial distribution of fluorescence within the membrane-bound regions of the flagella. The maximum intensity of kinesin-13mNG fluorescence was at the distal region of the flagellar tips and intensity sharply decreased within the first micrometer from the tip (*Figure 6D*). The quantification of kinesin-13mNG fluorescence intensity in the flagellar tip regions indicated significantly greater fluorescence intensity at the tips of caudal flagella compared to the longer anterior flagella (*Figure 6E*). However, the difference in intensity between the posteriolateral flagella and anterior flagella is not statistically significant (*Figure 6E*).

## Flagellar length control via length-dependent disassembly

From the above observations we developed a model of flagellar length control based on length-dependent disassembly (for mathematical details of this model see Materials and methods). A key aspect of the model is that the amount of kinesin-13 localized to the distal flagellar tip increases during de novo assembly and that accumulation of kinesin-13 at the tip is a consequence of its transport along the flagellum (*Figure 7A*). This leads to length-dependent disassembly of the flagella ($k^{-}_{S/L}$ in *Figure 7B*), to produce a stable length ($L_{S/L}^{*}$ in *Figure 7B*) when the disassembly rate equals the flagellar assembly rate ($k^{+}$ in *Figure 7B*). The assembly rate decreases linearly with length due to the depletion of the precursor (possibly tubulin) pool as the flagella elongate.

As a test of our model, we determined whether depletion of the tubulin precursor pool during ciliogenesis leads to a linear decrease in assembly rate with length ($k^{+}$ in *Figure 7B*). Quantitative measurements of de novo ciliogenesis are currently not technically feasible in *Giardia*. Therefore, to estimate the amount of precursor material available to each flagellum we treated trophozoites expressing mNeonGreen-tagged β-tubulin with 20 µM Taxol and measured the changes in flagellar length every hour (*Figure 8A*). We limited our analysis to five hours (the approximate doubling time of *Giardia* in culture), as Taxol has been demonstrated to induce mitotic spindle defects (*Sagolla et al., 2006*). We hypothesized that if the different flagellar pairs draw precursor material

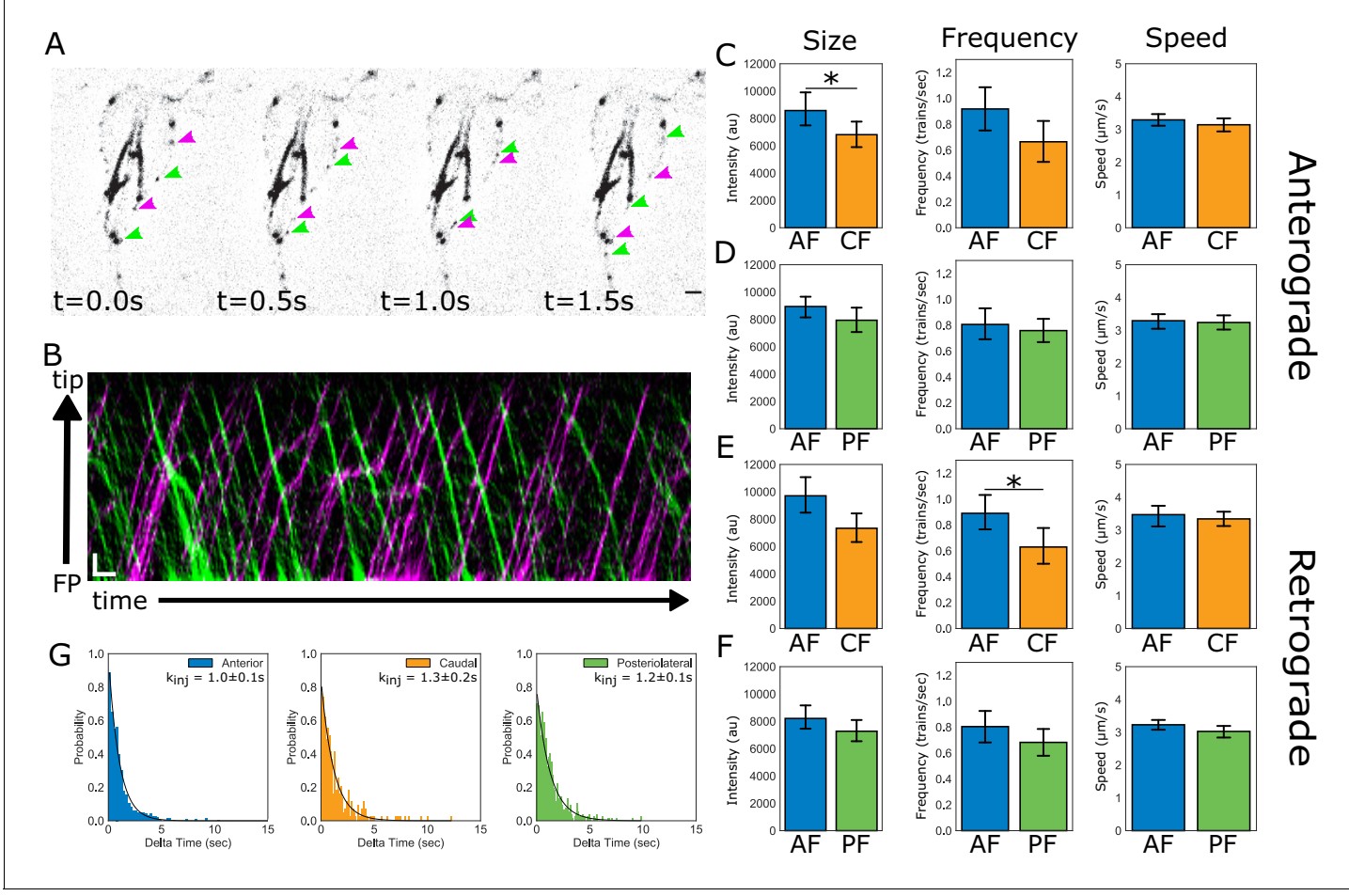

**Figure 4.** IFT dynamics are similar between flagellar pairs of different lengths. (**A**) Still images from time-lapse imaging of live trophozoites expressing IFT81mNG showing anterograde (magenta arrows) and retrograde IFT trains (green arrows). Scale bar, 2 μm. (**B**) A representative kymograph of IFT train trajectories within the membrane-bound anterior flagellum. Total time is ~26 s. Scale bar, 1 μm and 1 s. (**C**) Comparisons of anterograde IFT train intensity, frequency, and speed from anterior and caudal flagella. (**D**) Comparisons of anterograde IFT train intensity, frequency, and speed from anterior and posteriolateral flagella. (**E**) Comparisons of retrograde IFT train intensity, frequency, and speed from anterior and caudal flagella. (**F**) Comparisons of retrograde IFT train intensity, frequency, and speed from anterior and posteriolateral flagella. All plots show mean values with 95% confidence intervals. Student's t-test, *p<0.05. n = 22 cells for the anterior and caudal flagella, n = 42 cells for the anterior and posteriolateral flagella, from N = five independent experiments. (**G**) Frequency histograms of the time-lag between IFT train injections for anterior (blue), posteriolateral (green), and caudal (orange) flagella. Black line indicates a fit to a single exponential equation to measure the injection rate for each flagellar pair. Injection rates are indicated with 95% confidence intervals.

from common or equivalently sized pools, they would exhibit similar changes in flagellar length. Furthermore, small deviations in flagellar length at later time points would be indicative of the depletion of the pool. After five hours of Taxol treatment the anterior flagella increased by 4.1 ± 0.2 μm, the posteriolateral increased by 4.3 ± 0.3 μm, the ventral increased by 4.5 ± 0.2 μm, and the caudal increased by 5.4 ± 0.4 μm (*Figure 8B*). There were no significant differences in the elongation of the anterior, posteriolateral, or ventral flagella. However, the caudal flagella increased by a significantly greater amount than the other three pairs of flagella at the five-hour time point. There were also no significant differences in flagellar length between the four-hour timepoint and the five-hour time-point for any of the flagellar pairs.

Our model also predicts that the rate of spatial-decay of fluorescence away from the flagellar tip is length-independent (see Materials and methods). Specifically, the decay rate should be proportional to the diameter of the flagellum, a dimension that is consistent between all flagellar pairs. To determine the decay rate ($\lambda$) of the intensity profile within the first 1.2 μm from the flagellar tip, we plotted the fluorescence intensity of kinesin-13mNG and fit these data with the function

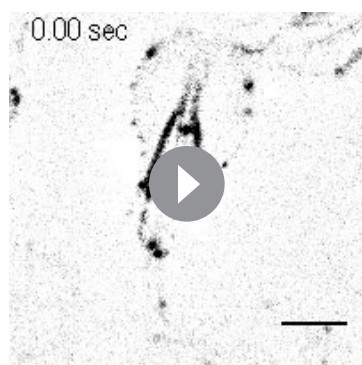

**Video 3.** Tracking IFT trains in *Giardia*. IFT train movement visualized using spinning-disc confocal microscopy in trophozoites expressing IFT81mNG. The video was recorded at ~13 frames/second and is played in real time (indicated in the top left corner, in seconds). Scale bar, 5 μm.
https://elifesciences.org/articles/48694#video3

$I(x) = I_0 \cosh\left(\frac{1.2 - x}{\lambda}\right)$ obtained from theory (see Materials and methods, *Figure 8E*). We found that the decay rate $(\lambda)$ is constant across all flagellar pairs (*Figure 8F*) and has a value on the order of the flagellum width. We also found that the majority of kinesin-13 is localized to the flagellar tip at equilibrium, with minimal fluorescence intensity measured along the length of the membrane-bound axoneme (*Figure 6D*, *Figure 8—figure supplement 1*).

Once a flagellum reaches its equilibrium length, our model predicts that the reservoir of kinesin-13 is exhausted and the majority of kinesin-13 is localized to the distal flagellar tip (*Figure 7A and C*). As a further test of our model of disassembly-dependent length control, we used Taxol to elongate the flagella for one hour and measured the fluorescence intensity of kinesin-13mNG at the distal flagellar tips (*Figure 8E*, *Figure 8—figure supplement 1*). The fluorescence intensity at the distal flagellar tips decreased slightly with Taxol-induced flagellar elongation in all measured flagella, but these differences are not statistically significant (*Figure 8D*). Consistent with length-independent decay, the rate of exponential decay of kinesin-13mNG with distance from the distal flagellar tip was unchanged in Taxol-elongated flagella (*Figure 8E and F*).

The exponential decay of fluorescence intensity from the distal flagellar tip suggests that kinesin-13 utilizes IFT transport only in the anterograde direction (*Figure 7C*). An important prediction of our model is that kinesin-13 will undergo turnover at the distal flagellar tip, but that this turnover is minimal, as the majority of kinesin-13 is localized there (*Figure 6D*). To test this prediction, we used FRAP and photobleached kinesin-13mNG at the tips of caudal, posteriolateral, and anterior flagella (*Figure 8G*, *Video 4*). The initial rates of recovery of kinesin-13 fluorescence were similar at the tips of all types of flagella analyzed (*Figure 8G and H*, and *Figure 8—figure supplement 2*). The maximum fluorescence recovery for all flagella was approximately 30% of the initial fluorescence intensity, representing a small mobile fraction of kinesin-13 in all flagella (*Figure 8H*, *Figure 8—figure supplement 2*).

## Discussion

*Giardia* trophozoites are a unique and ideal model to test the mechanisms of flagellar length control that enable multiciliated cells to possess flagella with different lengths. Like many model organisms, *Giardia* has canonical motile axonemes that are nucleated by basal bodies and have a conserved '9 +2' axoneme structure. *Giardia* also possesses the majority of IFT, BBSome, and motor proteins (kinesin-2, kinesin-13, and IFT dynein) that are essential components of flagellar length control mechanisms in diverse model systems (*Avidor-Reiss and Leroux, 2015*; *Lechtreck, 2015*). In contrast to other models, the eight *Giardia* axonemes are organized into four flagellar types with four different equilibrium lengths that include long, non-membrane-bound cytoplasmic regions (*Figure 1*). Distinct modes of flagellar beating of these different flagellar types are essential in *Giardia*'s life cycle for motility, cytokinesis, and excystation (*Buchel et al., 1987*; *Hardin et al., 2017*; *Lenaghan et al., 2011*).

Here we demonstrate two types of evolutionary innovations in both axonemal architecture and in flagellar length regulation that could be broadly generalizable to other eukaryotes. First, rather than accumulating at transition zone regions, IFT proteins accumulate and are injected into the membrane-bound regions as IFT trains at flagellar pore regions – the interface between the cytosol and compartmentalized membrane-bound axonemes. The second innovation is a modification of known flagellar length regulatory mechanisms that enables size control for multiple flagella of different lengths within the same cell. We suggest that the distinct lengths of *Giardia*'s different flagellar pairs

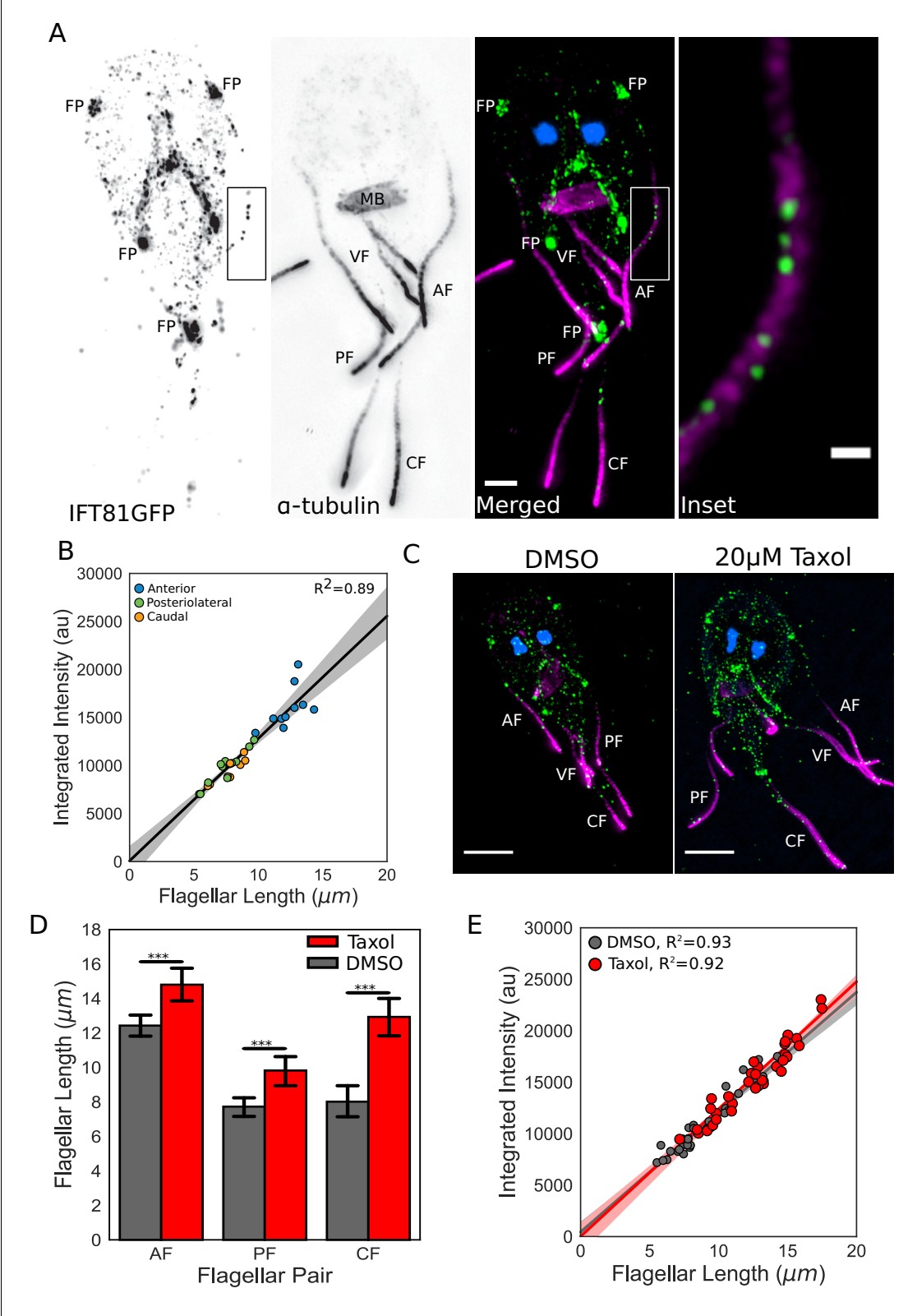

**Figure 5.** IFT injection is length-independent. (**A**) Representative structured illumination microscopy (SIM) image of a trophozoite expressing IFT81GFP (green) immunostained for α-tubulin (magenta) and stained with DAPI (blue). Scale bar, 2 μm. Boxed inset is enlarged on the right. Scale bar, 0.5 μm. (**B**) Total integrated intensity of IFT81GFP plotted versus flagellar length. Orange dots indicate caudal flagella, green dots indicate posteriolateral flagella, and blue dots indicate anterior flagella. Linear fit (black line) and coefficient of determination are indicated. Shading indicates 95% confidence

*Figure 5 continued on next page*

Figure 5 continued

interval. (C) Representative SIM images of IFT81GFP expressing trophozoites treated with DMSO (left) or 20 µM Taxol (right) for one hr, then fixed and stained as in A. (D) Flagellar lengths of IFT81GFP expressing trophozoites treated with DMSO (gray) or 20 µM Taxol (red). Ten cells from three separate experiments were measured for each condition. Student's t-test, ***p<0.001. (E) Total integrated intensity of IFT81GFP for trophozoites treated with DMSO (gray) or 20 µM Taxol (red) plotted versus flagellar length. Linear fit (gray and red lines) and coefficient of determination are indicated. Shading indicates 95% confidence interval.

are the result of length-independent IFT injection rates that are balanced by differential, length-dependent accumulation of the kinesin-13 at the distal flagellar tips.

## The flagellar pore acts as a diffusion barrier that is functionally analogous to the transition zone

Motor-IFT complexes mediate dynamic trafficking of structural and signaling proteins into the compartmentalized flagellum and are required for both flagellar assembly and maintenance (*Lechtreck, 2015*; *Reiter et al., 2012*). In many eukaryotes, IFT trains assemble and accumulate in the transition zone (TZ), a structurally and functionally distinct region where IFT proteins are compartmentalized and concentrated. At the TZ, anterograde IFT trains are loaded with axonemal structural material and transported to the distal flagellar tip by the heterotrimeric kinesin-2 complex (*Deane et al., 2001*; *Wingfield et al., 2017*). Retrograde trains are returned from the distal tip back to the cell body by cytoplasmic dynein (*Pazour et al., 1999*). Regulation of IFT train assembly, regulation of IFT motors and, ultimately, regulation of flagellar length, are commonly attributed to regulatory proteins localizing to the TZ (*Cole et al., 1998*; *Wei et al., 2013*; *Wingfield et al., 2017*).

*Giardia*'s unique flagellar structure and the lack of a canonical TZ raises questions as to where IFT particles assemble, mature, and are injected into the membrane-bound axonemal regions (*Avidor-Reiss and Leroux, 2015*; *Barker et al., 2014*). Prior analyses localized both kinesin-2 homologs and two IFT proteins not only to *Giardia*'s membrane-bound flagella, but also to the cytoplasmic regions of each axoneme (*Dawson et al., 2007*; *Hoeng et al., 2008*). By tagging and imaging most of *Giardia*'s IFT proteins in this study, we confirmed that the majority of anterograde and retrograde IFT proteins localize to the membrane bound regions and to the cytoplasmic regions (*Figure 2*). Rather than at basal bodies, tagged IFT proteins concentrate at each of the eight flagellar pores (*Figure 2A* and *Hoeng et al., 2008*). Anterograde kinesin-2a and kinesin-2b motors also accumulate at each of the flagellar pores and distal tips, yet they lack a similar intensity of localization to the cytoplasmic regions of axonemes or basal bodies as the IFT proteins (*Figure 2*).

In addition to IFT-mediated trafficking on membrane-bound axonemes (*Figure 4*), the behavior and turnover of IFT proteins in cytoplasmic regions and at pores is dynamic (*Figure 3A,B* and *Videos 1* and *2*). Notably, the turnover of IFT proteins at the pores is about three times faster than in the cytoplasmic regions (*Figure 3C,D*). The discrepancy in IFT turnover rates between the flagellar pores and the cytoplasmic regions implies that flagellar pore accumulations of IFT proteins are not solely due to diffusive exchange with the cytoplasmic axonemes. Both photobleaching of IFT in regions of the axonemes (*Figure 3C,D*) and modelling (*Figure 3F*) support that the accumulation of IFT proteins at the flagellar pores (*Figure 2C*) is driven by the combination of both diffusive and directed transport of IFT proteins. Thus, we contend that the eight flagellar pore complexes are functionally analogous to the transition zone, concentrating IFT proteins and acting as a diffusion barrier between the cytoplasmic and membrane-bound axoneme compartment. Furthermore, this work supports that the flagellar pores are the sites of IFT injection into the membrane-bound axonemes. Though we currently lack direct spatial resolution of IFT particle maturation in *Giardia*, we speculate that immature IFT particles first associate with cytoplasmic axonemal regions and mature at the eight flagellar pores, prior to their injection into the compartmentalized, membrane-bound axonemal region. The injection mechanism is likely analogous to that of other flagellates, yet in the absence of a TZ, *Giardia* must employ alternative or novel components.

*Giardia*'s use of flagellar pore complexes rather than transition zone proteins in IFT calls into question the necessity of the TZ complex for IFT-mediated assembly in other flagellated cell types. Indeed, work in *C. elegans* supports that TZ function may be independent from IFT, as mutations in TZ complex proteins do not adversely impact IFT-mediated trafficking (*Williams et al., 2011*). Further characterization of the ultrastructure and the composition of the flagellar pore complex in

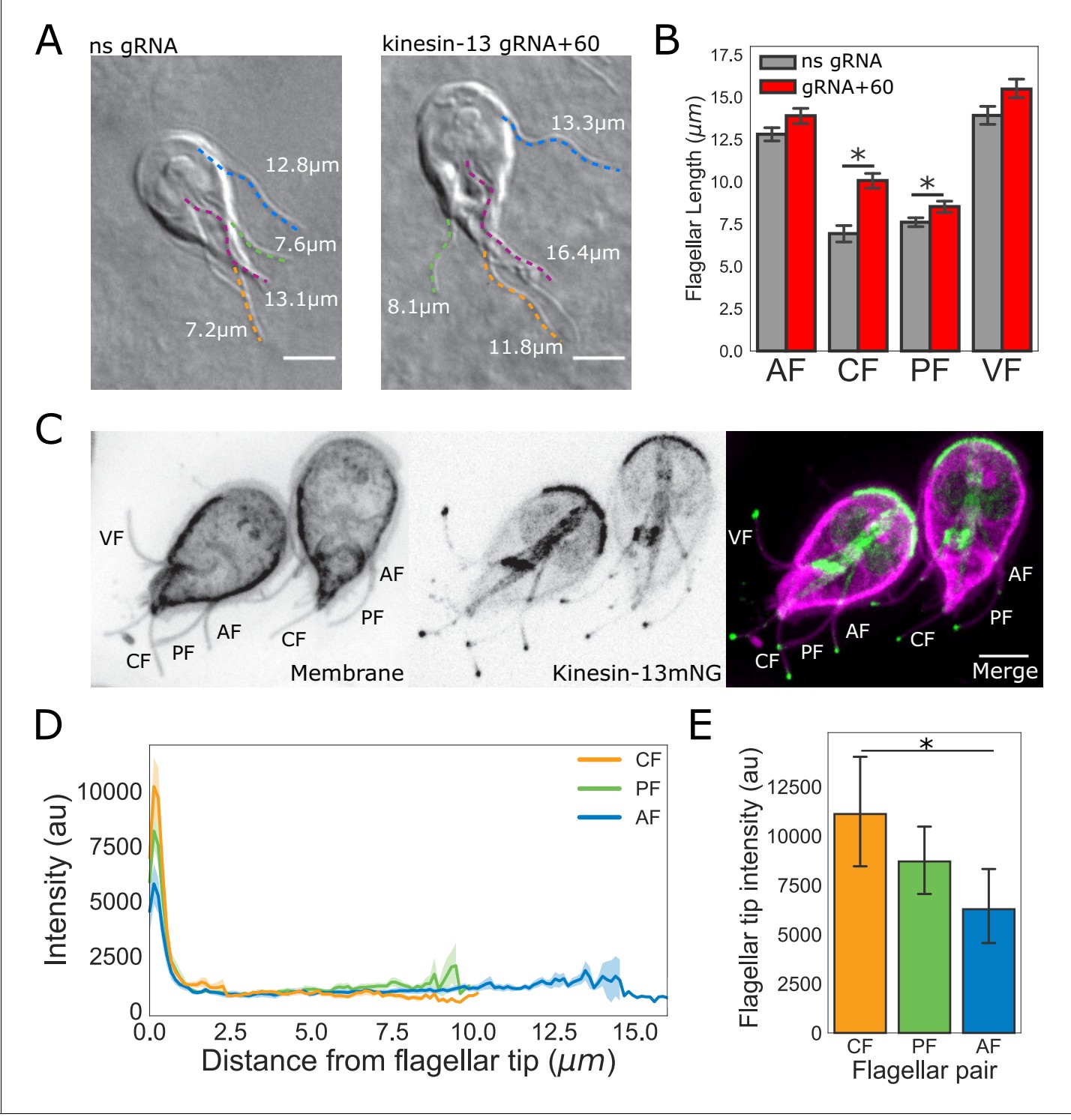

**Figure 6.** The intensity of kinesin-13 flagellar tip localization is inversely correlated with flagellar length. (**A**) Representative images and (**B**) quantification of CRISPRi mediated knockdown of kinesin-13 (gRNA+60, red) as compared to a non-specific (ns, gray) gRNA. Blue traces indicate anterior flagella, magenta traces indicate the ventral flagella, green traces indicate the posteriolateral flagella, and orange traces indicate the caudal flagella. n ≥ 30 cells from two separate experiments were measured for each condition. Means and 95% confidence intervals are indicated. (**C**) Representative image of trophozoites expressing kinesin-13mNG with the cell membrane labeled to indicate the membrane-bound regions of the flagella. Scale bar, 5 μm. (**D**) Kinesin-13mNG intensity profiles from the flagellar tip to the base of the membrane-bound regions of caudal (orange), posteriolateral (green), and anterior (blue) flagella. Shading indicates standard error of the mean. n ≥ 23 for each flagellar pair, from two independent experiments. (**E**) Mean flagellar tip intensity plotted for each flagellar pair. 95% confidence intervals are indicated. Student's t-test, *p<0.05.

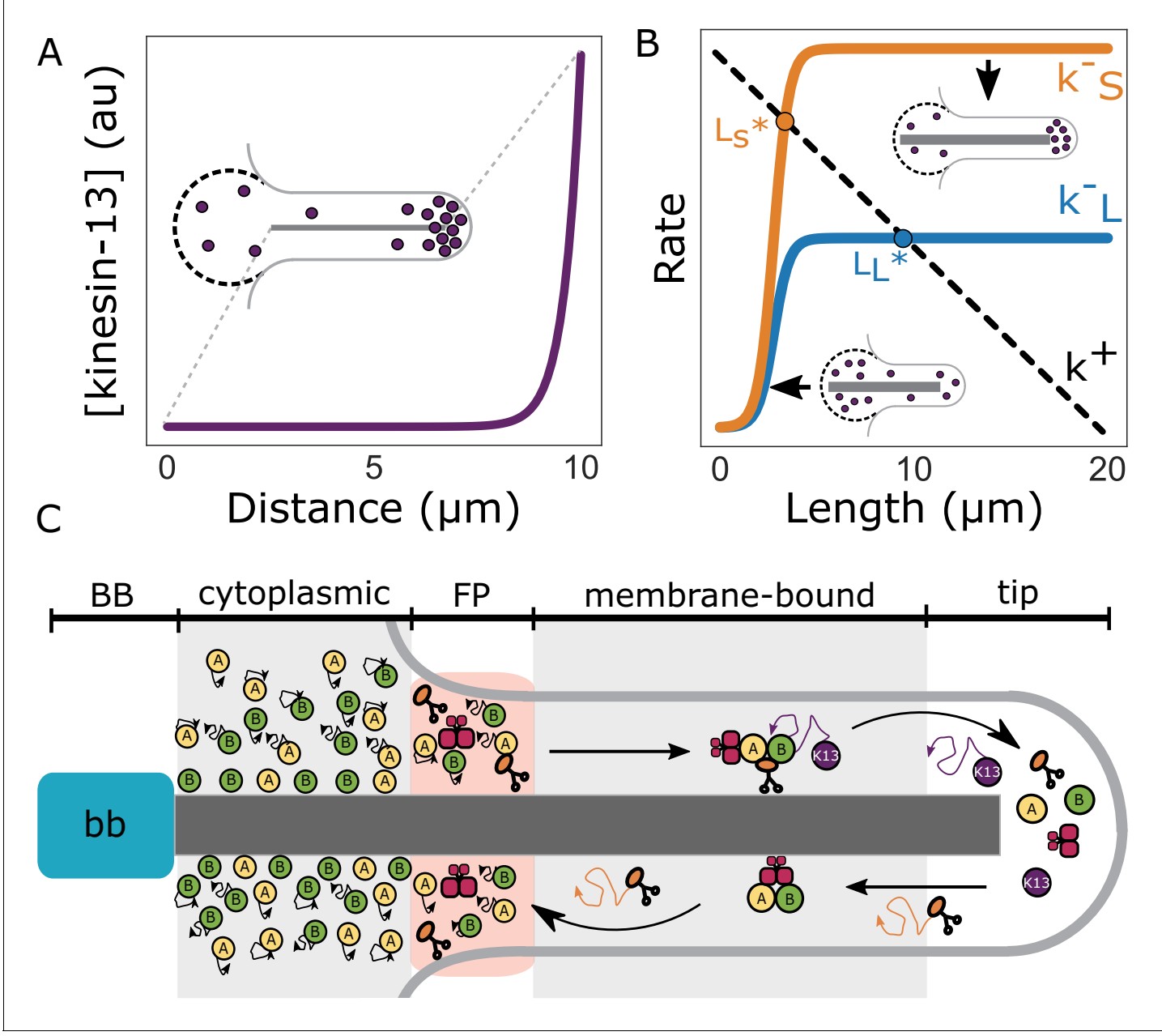

**Figure 7.** Length-dependent disassembly controls flagellar length. (**A**) Theoretically predicted concentration profile of kinesin-13 (purple) at equilibrium flagellar length. Inset depicts a schematic representation of kinesin-13 concentration at the flagellar tip and in the reservoir (dashed lines). (**B**) Theoretically predicted disassembly rates for short ($k^-_S$) and long ($k^-_L$) flagella as a function of length (Materials and methods). Dashed line indicates the proposed rate of assembly ($k^+$). Intersections of disassembly and assembly rates generate two distinct equilibrium flagellar lengths ($L_S^*$, $L_L^*$). (**C**) Schematic of flagellar assembly and maintenance in *Giardia lamblia*. IFT particles (yellow and green circles) move diffusively in the cytoplasmic axoneme regions. IFT trains are assembled in the flagellar pore region and are injected into the membrane-bound region of the axoneme. Within the membrane-bound region, IFT particles undergo anterograde transport via kinesin-2 (orange) mediated transport until they reach the distal flagellar tip. IFT trains are reorganized into retrograde directed trains and carried back to the flagellar base by IFT dynein (red). Kinesin-2 and kinesin-13 (purple) are not included in retrograde IFT trains, and instead diffuse back to the flagellar base. While kinesin-2 can freely diffuse to the flagellar base, kinesin-13 can be 'recaptured' by anterograde IFT trains and carried back to the distal tip. Unidirectional transport coupled with free diffusion is expected to give a profile that decreases linearly from the flagellar tip to the base (Kinesin-2). Unidirectional transport coupled with diffusion and anterograde recapture gives a profile that decreases exponentially from the distal flagellar tip (Kinesin-13) (*Naoz et al., 2008*).

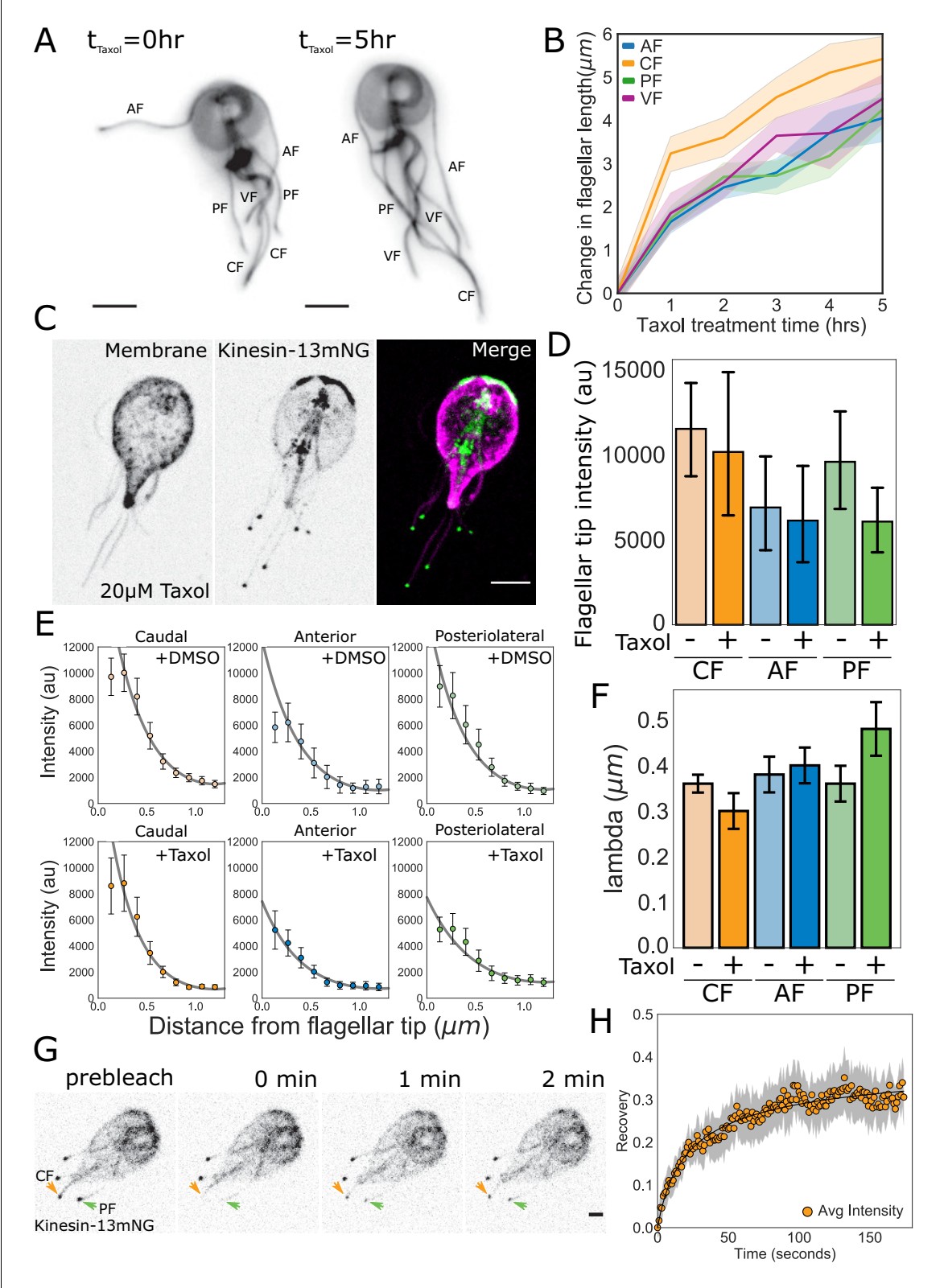

**Figure 8.** Quantitative tests of specific predictions from the disassembly-dependent flagellar length control model in *Giardia*. (**A**) Representative images and (**B**) quantification of flagellar length changes in trophozoites expressing mNG-β-tubulin and treated with 20 μM Taxol for 5 hr. Flagellar pairs are indicated. Scale bar, 5 μm. n ≥ 35 cells from two separate experiments were measured for each time-point. Means and 95% confidence intervals are indicated. (**C**) Representative image of trophozoites expressing kinesin-13mNG and treated with 20 μM Taxol for 1 hr, with the cell

*Figure 8 continued*

membrane labeled to indicate the membrane-bound regions of the flagella. Scale bar, 5 µm. (D) Flagellar tip intensity of kinesin-13mNG expressing trophozoites treated with DMSO ('-') or 20 µM Taxol ('+'). (E) Kinesin-13mNG fluorescence intensity decay within the first 1.2 µm of the flagellar tip of caudal, anterior, and posteriolateral flagella trophozoites treated with DMSO (top panel) or 20 µM Taxol (bottom panel). Means and 95% confidence interval are indicated. Gray lines indicate fits to obtain the decay rate ($\lambda$) of the intensity profile. (F) Mean decay rate (lambda) for caudal (CF), anterior (AF), and posteriolateral (PF) flagella treated with DMSO ('-') or 20 µM Taxol ('+'). n $\geq$ 12 cells from two separate experiments were measured for each condition. Means and 95% confidence intervals are indicated. (G) Time series images of trophozoites expressing kinesin-13mNG prebleach, immediately post-bleach (0 min, arrows), and during recovery (time in minutes) for caudal and posteriolateral flagellar tip regions. Scale bar, 2 µm. (H) Time averaged fluorescent recovery of caudal flagellar tip regions following photobleaching. Solid black lines indicate fit of the entire recovery phase and shading indicates the 95% confidence interval. n = 19 caudal flagellar tips, from two independent experiments.

The online version of this article includes the following figure supplement(s) for figure 8:

**Figure supplement 1.** Kinesin-13mNG flagellar length changes and intensity profiles following flagellar elongation with Taxol.
**Figure supplement 2.** Kinesin-13mNG fluorescence recovery after photobleaching of anterior, caudal, and posteriolateral flagella.

*Giardia* will be key to determining how this complex mediates compartmentalization and regulates IFT injection.

Lastly, the mode of flagellar inheritance and IFT-mediated assembly during cytokinesis may also impact the assembly of the cytoplasmic or membrane-bound regions of each *Giardia* flagellum. Cytoplasmic regions of the caudal and anterior axonemes of each daughter are proposed to be structurally inherited from the parental cell, whereas posteriolateral and ventral flagellar pairs are assembled de novo during cytokinesis (*Hardin et al., 2017*; *Nohynková et al., 2006*; *Sagolla et al., 2006*). This combination of structurally inherited flagella and both cytosolic and compartmentalized ciliogenesis in *Giardia* creates spatially distinct regions that are not diffusion-limited. While membrane-bound regions are assembled using IFT-mediated mechanisms (*Hoeng et al., 2008*; *McInally et al., 2019*), the cytoplasmic regions of each axoneme may be assembled in an IFT-independent manner. Neither kinesin-2 knockdown nor the expression of a dominant negative kinesin-2 affect cytoplasmic axoneme length (*Hoeng et al., 2008*; *McInally et al., 2019*). While we have not directly imaged the assembly of the cytoplasmic axonemal regions in this study, the lack of active transport of IFT proteins on cytoplasmic regions supports an IFT-independent mechanism for the de novo assembly of the posteriolateral and ventral flagellar pairs (*Hardin et al., 2017*; *Sagolla et al., 2006*). Flagellar pore complexes may also be required to direct the cytoplasmic axonemes to their defined exit points at the flagellar pores during cell division (*McInally and Dawson, 2016*).

The necessity of cytoplasmic ciliogenesis and inheritance to precede the assembly of the membrane-bound regions during cytokinesis may make the TZ dispensable in *Giardia.* Atypical modes of ciliogenesis are not unprecedented, as the apicomplexans *Plasmodium* and *Toxoplasma* also lack IFT, BBsome, or TZ components and use IFT-independent or compartment-independent mechanisms to assemble flagella with no membrane invagination or basal body migration (*Avidor-Reiss and Leroux, 2015*; *Barker et al., 2014*; *Briggs et al., 2004*). Both mammalian and *Drosophila* sperm flagella also employ variations of cytoplasmic ciliogenesis that require the invagination of the basal body into the cytoplasm following the initiation of compartmentalized ciliogenesis (*Avidor-Reiss et al., 2017*).

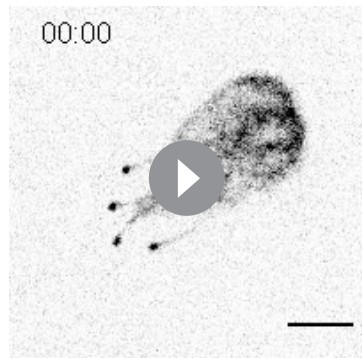

**Video 4.** Fluorescence recovery of kinesin-13mNG after photobleaching of caudal and posteriolateral flagellar tips. Fluorescence recovery following photobleaching of the flagellar tips of caudal (left) and posteriolateral (right) flagellar tips in trophozoites expressing kinesin-13mNG. The video was recorded at one frame/second and is played at 10x increased speed. Time post-bleach (in minutes) is indicated in the top left corner. Scale bar, 5 µm.

https://elifesciences.org/articles/48694#video4

## Flagellar pairs of different lengths have similar IFT train injection rates

Length-dependent assembly and/or length-dependent disassembly are required to maintain the equilibrium length of a flagellum (*Mohapatra et al., 2016*). As described in the 'balance-point' model of flagellar length control, the rate of subunit addition must be balanced by the rate of sub-unit removal to set the length of a dynamic structure at a specific size (*Marshall et al., 2005*; *Mohapatra et al., 2016*). Flagellar assembly occurs with deceleratory kinetics and the length-dependence of the assembly rate is thought to be due to a limited amount of IFT material to mediate assembly (*Engel et al., 2009*; *Mohapatra et al., 2016*; *Rosenbaum and Child, 1967*; *Tamm, 1967*). The flagellar assembly rate is thus a result of IFT train injection into the membrane-bound region of the flagellum. Length-dependent assembly is proposed to arise from the depletion of the assembly motor, kinesin-2, at the flagellar base and the diffusive return of this essential IFT component from the flagella tip (*Chien et al., 2017*; *Fai et al., 2019*; *Hendel et al., 2018*). In contrast, the disassem-bly rate is thought to be length-independent (*Engel et al., 2009*; *Ludington et al., 2013*). In this way, the amount of kinesin-2 available to be incorporated into IFT trains acts as a length-ruler of flagella.

*Giardia's* eight flagella of four different equilibrium lengths (*Figure 1*) pose a challenge to the canonical model of flagellar length control. Based on flagellar length control paradigms developed in other flagellated protists, one might predict that *Giardia* differentially regulates flagellar assembly between flagella of different equilibrium lengths by tuning specific aspects of IFT dynamics (particle size, number, or injection frequency). Yet, we show here that IFT dynamics and IFT train injection rates are consistent between flagella (*Figures 4* and *5*). Furthermore, the injection rate remained constant when increased flagellar length was pharmacologically induced by Taxol treatment (*Figure 5E*). These observations imply that tuning of assembly rates is not a regulatory mechanism used by *Giardia*. The length-independent IFT injection observed in *Giardia* contrasts with observa-tions in the green alga *Chlamydomonas*, where IFT train injection decreases with increasing flagellar length to provide a length-dependent assembly rate (*Chien et al., 2017*; *Hendel et al., 2018*). Length-independent injection of IFT trains has recently been observed in *Trypanosoma* and a 'grow-and-lock' model has been proposed wherein the flagellum grows at a constant rate until it is locked into a stable length (*Bertiaux et al., 2018*). While we also observe length-independent IFT injection, the dynamic nature of *Giardia's* flagella via pharmacological and genetic methods suggest that a locking mechanism is unlikely to explain the maintenance of multiple flagellar pairs of distinct lengths. Several kinases are known to regulate either assembly or disassembly rates in *Tetrahymena* and *Chlamydomonas* (CALK, LF4, Nrks/Neks), and *Giardia* has 198 Nek proteins whose functions are yet to be determined (*Berman et al., 2003*; *Bradley and Quarmby, 2005*; *Hilton et al., 2013*; *Manning et al., 2011*; *Meng and Pan, 2016*; *Wloga et al., 2006*). It is possible that these uncharac-terized Nek proteins function to regulate specific aspects of cytoskeletal organization in *Giardia*, including flagellar length.

## Length-dependent kinesin-13 mediated disassembly maintains flagella of unique lengths

*Giardia* has a single kinesin-13 homolog that regulates length of axonemes as well as the dynamics of various MT arrays (e.g., two spindles and the median body) as the ectopic expression of a domi-nant-negative, kinesin-13 rigor mutant or CRISPRi-mediated knockdown of kinesin-13 in *Giardia* (*Fig-ure 6*) results in dramatically longer flagella (*Dawson et al., 2007*; *McInally et al., 2019*). Kinesin-13 contributions to both IFT-mediated flagellar assembly and disassembly have also been investigated in other microbial flagellates such as *Leishmania, Trypanosoma, Tetrahymena,* and *Chlamydomonas* (*Blaineau et al., 2007*; *Chan and Ersfeld, 2010*; *Piao et al., 2009*; *Vasudevan et al., 2015*; *Wang et al., 2013*). In *Leishmania*, overexpression of one (of the five) kinesin-13 homologs promotes decreased flagellar length, whereas the knockdown of the orthologous kinesin-13 in *Trypanosoma brucei* promotes increased flagellar length (*Blaineau et al., 2007*; *Chan and Ersfeld, 2010*). In the green alga *Chlamydomonas*, kinesin-13 is transported to the distal flagellar tip via IFT during the induction of flagellar resorption. Depletion of the sole kinesin-13 in *Chlamydomonas* results in shorter flagella due to the depletion of the cytoplasmic tubulin pool required for IFT-mediated assembly (*Piao et al., 2009*; *Wang et al., 2013*). Lastly, in *Tetrahymena*, cell body MTs are short-ened by kinesin-13, but this activity is not required for liberating ciliary precursor tubulin.

In the absence of differences in length-dependent axoneme assembly in *Giardia,* we investigated the contribution of kinesin-13 mediated flagellar disassembly to flagellar length control. Generally, kinesin-13 depolymerizes the ends of microtubules via ATP hydrolysis and undergoes one dimensional diffusion along the lattice with no preference for either the plus or minus-end of the microtubule (*Cooper and Schafer, 2000*; *Desai et al., 1999*; *Hunter et al., 2003*; *Helenius et al., 2006*). Here we determined that kinesin-13 differentially and dynamically localizes to the distal flagellar tip in flagellar pairs of different lengths (*Figure 7*). This observation is consistent with kinesin-13 depolymerizing axonemal MTs in a length-dependent manner. In the absence of other active transport mechanisms within the membrane-bound flagellum, we propose that kinesin-13 is a cargo of IFT (*Piao et al., 2009*). Due to turnover of kinesin-13 at the distal flagellar tip, coupled with the apparent exponential distribution of kinesin-13 fluorescence intensity within the flagellum (*Figure 8*), we suggest that the kinesin-13 mediated disassembly activity at the distal tips is a primary driver of differential flagellar length regulation in *Giardia*.

In addition to a direct role in MT disassembly at the flagellar tips, it is possible that kinesin-13 activity indirectly affects IFT-mediated assembly by regulating cytoplasmic tubulin pools through liberation of tubulin subunits from the median body (*Dawson et al., 2007*). The median body (*Figure 1*, 'MB') is a semi-organized interphase MT array that has been proposed to act as a cytoplasmic reservoir of tubulin prior to cell division (*Hardin et al., 2017*) and overexpression of a rigor mutant kinesin-13 affected median body volume as well as flagellar lengths (*Dawson et al., 2007*).

## A disassembly mediated model for flagellar length control

We propose a model where *Giardia* controls flagellar length through the modulation of axonemal-specific, length-dependent flagellar disassembly rates, rather than length-dependent IFT-mediated assembly rates as reported in other systems. In this model the four equilibrium flagellar lengths are achieved by modulating the amount of kinesin-13 localized to the distal flagellar tip during assembly (*Figure 7*); however, it remains unclear how kinesin-13 is differentially transported or regulated at the eight different flagellar tips.

We propose that during flagellar assembly, kinesin-13 is transported to the flagellar tips from a flagellum-specific reservoir (possibly the flagellar pores) via anterograde IFT (*Figure 7C*). As the flagellum elongates, kinesin-13 is depleted from this reservoir while accumulating at the tip (*Figure 7A* inset). The concentration of kinesin-13 at the distal tip increases until it reaches the critical concentration at which point the kinesin-13-mediated disassembly balances the assembly rate (*Figure 7B*). A key assumption of our model is that the kinesin-13 at the flagellar tip can bind to the axoneme to induce microtubule depolymerization. If so, the fluorescence intensity at the flagellar tip is a direct readout of the depolymerizing activity of kinesin-13. Our observation that the tips of shorter (caudal) flagella exhibit more kinesin-13mNG fluorescence than longer (anterior) flagella is consistent with this idea (*Figure 6D and E*).

The sharp decrease in fluorescence intensity immediately proximal to the flagellar tip suggests that kinesin-13 does not undergo directed retrograde transport on the axoneme, but rather diffuses from the flagella tip toward the base (*Figure 7C*). Furthermore, this exponential decay suggests that during diffusion from the flagellar tip, kinesin-13 is recaptured by anterograde IFT trains, sequestering it to the tip region (*Naoz et al., 2008*) (*Figure 7C*). These aspects of our model remain to be directly tested, but they are supported by the length-independent decay rate of kinesin-13mNG, both at equilibrium and during elongation with Taxol (*Figure 8F*). Furthermore, we observe minimal recovery of kinesin-13 at the flagellar tip following photobleaching indicating that there is small mobile fraction restricted to the flagellar tip (*Figure 8H*). We have been unable to directly observe kinesin-13 undergoing transport within the membrane-bound flagellum, however our model predicts—and our measurements show— that at equilibrium the majority of kinesin-13 is localized to the distal flagellar tip (*Figure 8D and E*). We expect that the development of methods that permit the imaging of flagella undergoing regeneration or de novo assembly will allow us to directly test this aspect of our model. Furthermore, the development of new genetic manipulation strategies in *Giardia* will help to identify other possible regulators of flagellar length control.

## Control of flagellar length by a limiting precursor pool

While our experimental observations are consistent with our model of disassembly-dependent flagellar length control, we cannot rule out the possibility that length control is achieved through an assembly-dependent mechanism. Although we do not observe differences in IFT injection or transport between the different flagellar pairs, it is still possible that there are other limiting components to flagellar assembly. In this case, we would expect that this limiting precursor component would be depleted from the pool during ciliogenesis until it is balanced by the flagellum specific disassembly rate imparted by kinesin-13 at the flagellar tip. Although we are unable to directly measure the precursor pool size for each flagellum, our comparisons of flagellar length during long exposures to Taxol indicate that the precursor pool for each flagellum is depleted during ciliogenesis. Each flagellum likely draws precursor material from either a common pool or pools of equivalent sizes (*Figure 8B*). Importantly, this depletion contrasts with the length-dependent assembly rate in *Chlamydomonas* that is due to the exhaustion of IFT material at the flagellar base (*Chien et al., 2017*; *Hendel et al., 2018*). In the absence of length-dependent IFT injection, we propose that the length-dependent assembly rate could be due the depletion of the structural precursor pool. This suggests that *Giardia* may employ a variation of the 'differential cargo loading model', wherein the loading of axoneme structural cargo onto IFT trains is length-dependent (*Craft et al., 2015*; *Wren et al., 2013*). A key difference to this model is that in *Giardia*, IFT injection remains constant as the precursor pool becomes depleted. Therefore, while the flux of structural material toward the flagellar tip in *Chlamydomonas* may be regulated by both length-dependent IFT injection and the cargo capacity of IFT trains, we expect that in *Giardia* the precursor pool size would be the primary driver of the assembly rate. With further development and adaptation of advanced single molecule imaging techniques in *Giardia*, these aspects of our model can be directly tested.

Between species and even within cell types of the same species, eukaryotic cells have diverse cytoskeletal architectures that enable innovations in motility and other cellular functions. Defining the molecular mechanisms by which non-canonical flagellated cells like *Giardia* alter the balance of well-studied IFT assembly and kinesin-13 mediated disassembly mechanisms illuminates how cells can evolve varied morphological forms. Beyond microbial flagellates, we expect the mechanistic and structural innovations we describe in *Giardia* will echo well-described variations in flagellar structure, type, and number found in different cell types in humans and other multicellular model systems.

## Materials and methods

**Key resources table**

| Reagent type (species) or resource | Designation | Source or reference | Identifiers | Additional information |
|---|---|---|---|---|
| Genetic reagent (*Giardia lamblia*) | BBS2 | GiardiaDB | RRID:SCR_013377 GL50803_23934 | C-terminal GFP tagged cell line maintained in Dawson Lab |
| Genetic reagent (*Giardia lamblia*) | BBS4 | GiardiaDB | RRID:SCR_013377 GL50803_10529 | C-terminal GFP tagged cell line maintained in Dawson Lab |
| Genetic reagent (*Giardia lamblia*) | BBS5 | GiardiaDB | RRID:SCR_013377 GL50803_8146 | C-terminal GFP tagged cell line maintained in Dawson Lab |
| Genetic reagent (*Giardia lamblia*) | Beta-tubulin | GiardiaDB | RRID:SCR_013377 GL50803_101291 | N-terminal mNG tagged cell line maintained in Dawson Lab |
| Genetic reagent (*Giardia lamblia*) | IFT 38 | GiardiaDB | RRID:SCR_013377 GL50803_16707 | C-terminal GFP tagged cell line maintained in Dawson Lab |
| Genetic reagent (*Giardia lamblia*) | IFT 54 | GiardiaDB | RRID:SCR_013377 GL50803_9098 | C-terminal GFP tagged cell line maintained in Dawson Lab |

*Continued on next page*

Continued

| Reagent type (species) or resource | Designation | Source or reference | Identifiers | Additional information |
|---|---|---|---|---|
| Genetic reagent (*Giardia lamblia*) | IFT 56 | GiardiaDB | RRID:SCR_013377 GL50803_16375 | C-terminal GFP tagged cell line maintained in Dawson Lab |
| Genetic reagent (*Giardia lamblia*) | IFT 57 | GiardiaDB | RRID:SCR_013377 GL50803_14713 | C-terminal GFP tagged cell line maintained in Dawson Lab |
| Genetic reagent (*Giardia lamblia*) | IFT 74/72 | GiardiaDB | RRID:SCR_013377 GL50803_9750 | C-terminal GFP tagged cell line maintained in Dawson Lab |
| Genetic reagent (*Giardia lamblia*) | IFT 80 | GiardiaDB | RRID:SCR_013377 GL50803_17223 | C-terminal GFP tagged cell line maintained in Dawson Lab |
| Genetic reagent (*Giardia lamblia*) | IFT 81 | GiardiaDB | RRID:SCR_013377 GL50803_15428 | C-terminal GFP and mNG tagged cell line (episomal and integrated) maintained in Dawson Lab |
| Genetic reagent (*Giardia lamblia*) | IFT 88 | GiardiaDB | RRID:SCR_013377 GL50803_16660 | C-terminal GFP tagged cell line maintained in Dawson Lab |
| Genetic reagent (*Giardia lamblia*) | IFT 121 | GiardiaDB | RRID:SCR_013377 GL50803_87817 | C-terminal GFP tagged cell line maintained in Dawson Lab |
| Genetic reagent (*Giardia lamblia*) | IFT 122 | GiardiaDB | RRID:SCR_013377 GL50803_16547 | C-terminal GFP tagged cell line maintained in Dawson Lab |
| Genetic reagent (*Giardia lamblia*) | IFT 140 | GiardiaDB | RRID:SCR_013377 GL50803_17251 | C-terminal GFP tagged cell line maintained in Dawson Lab |
| Genetic reagent (*Giardia lamblia*) | IFT 172 | GiardiaDB | RRID:SCR_013377 GL50803_17105 | C-terminal GFP tagged cell line maintained in Dawson Lab |
| Genetic reagent (*Giardia lamblia*) | Kinesin-13 | GiardiaDB | RRID:SCR_013377 GL50803_16945 | C-terminal mNG tagged cell line maintained in Dawson Lab |
| Genetic reagent (*Giardia lamblia*) | Kinesin-2a | GiardiaDB | RRID:SCR_013377 GL50803_17333 | C-terminal GFP tagged cell line maintained in Dawson Lab |
| Genetic reagent (*Giardia lamblia*) | Kinesin-2b | GiardiaDB | RRID:SCR_013377 GL50803_16456 | C-terminal mNG tagged cell line maintained in Dawson Lab |
| Strain, strain background (*Giardia lamblia*) | WBC6 | ATCC | ATCC 50803 | Cell line maintained in Dawson Lab |
| Recombinant DNA reagent | pKS_mNeon Green-N11_NEO | https://doi.org/10.1073/pnas.1705096114 | | |
| Recombinant DNA reagent | pKS_mNeon Green-N11_PAC | https://doi.org/10.1073/pnas.1705096114 | | |
| Recombinant DNA reagent | *Giardia* Gateway cloning destination vector pcGFP1Fpac | https://doi.org/10.1016/S0091-679X(10)97017-9 | GenBank MH048881.1 | |
| Antibody | TAT-1 (mouse monoclonal) | Sigma-Aldrich | Anti-α-Tubulin, 00020911 | IF: (1:250) |
| Antibody | anti-GFP (rabbit polyclonal) | Sigma-Aldrich | | IF: (1:500) |
| Software, algorithm | KymographClear 2.0 | https://doi.org/10.1091/mbc.e15-06-0404 | | |

*Continued on next page*

*Continued*

| Reagent type (species) or resource | Designation | Source or reference | Identifiers | Additional information |
|---|---|---|---|---|
| Software, algorithm | KymographDirect | https://doi.org/10.1091/mbc.e15-06-0404 | | |
| Chemical compound, drug | Taxol | Sigma-Aldrich | T7402 | Concentration (in DMSO): 20 µM |

## Strains and culture conditions

*Giardia lamblia* (ATCC 50803) strains were cultured in modified TYI-S-33 medium supplemented with bovine bile and 5% adult and 5% fetal bovine serum in sterile 16 ml screw-capped disposable tubes (BD Falcon). Cultures were incubated upright at 37°C without shaking as previously described (*Hagen et al., 2011*). GFP and mNeonGreen-tagged IFT strains were created by electroporation of episomal vectors into strain WBC6 using approximately 20 µg plasmid DNA (*Hagen et al., 2011*). Tagged strains were maintained with antibiotic selection (50 µg/ml puromycin and/or 600 µg/ml G418) (*Hagen et al., 2011*). Trophozoites treated with Taxol were grown to confluency and split into 6 ml culture tubes 2 hr prior to incubation with Taxol (20 µM final concentration) or DMSO (0.2% final concentration) for the times indicated. Live or fixed imaging was performed as described below.

## Construction of episomal and integrated C-terminal GFP and mNeonGreen-tagged strains

All intraflagellar transport (IFT), BBSome, kinesin-2, and kinesin-13 homologs were identified by homology searches in the *Giardia* genome (GL50803) using GiardiaDB (*Aurrecoechea et al., 2009*). C-terminal GFP tagged constructs were created by PCR amplification of genomic DNA and subsequent cloning of the amplicons into a *Giardia* Gateway cloning vector (*Hagen et al., 2011*). mNeonGreen tagged strains for live imaging of beta-tubulin, IFT81, kinesin-2b, and kinesin-13 were constructed by PCR amplification of beta-tubulin (GL50803_101291), IFT81 (GL50803_15428), kinesin-2b (GL50803_16456), and kinesin-13 (GL50803_16945) genes from genomic DNA using primers designed by the NEBuilder Assembly Tool (New England Biolabs) for Gibson assembly (*Gibson et al., 2009*). The resulting amplicons were gel purified using a Zymoclean Gel DNA Recovery kit (Zymo Research) and cloned into pKS_mNeonGreen-N11_PAC (*Hardin et al., 2017*; *Shaner et al., 2013*) using Gibson assembly. The resulting mNeonGreen-beta-tubulin, IFT81mNeon-Green, and IFT81GFP plasmids were linearized via restriction digest for integration into the native locus (*Gourguechon and Cande, 2011*). To verify integration, total genomic DNA was extracted from tagged IFT81 strains using DNA STAT-60 (Tel-Test, Inc), and integration of the C-terminal tag was confirmed by PCR amplification (*Figure 2—figure supplement 1*).

## Immunostaining and light microscopy

*Giardia* trophozoites were grown to confluency as described above. Media was then replaced with 1x HBS (37°C), and the cultures were incubated at 37°C for 30 min. To detach and harvest cells, culture tubes were incubated on ice for 15 min and centrifuged at 900 x g, 4°C for five minutes. Pellets were washed twice with 6 ml cold 1x HBS and resuspended in 500 µl 1x HBS. Cells (250 µl) were attached to warm coverslips (37°C, 20 min), fixed in 4% paraformaldehyde, pH 7.4 (37°C, 15 min), washed three times with 2 ml PEM, pH 6.9 (*Woessner and Dawson, 2012*) and incubated in 0.125M glycine (15 min, 25°C) to quench background fluorescence. Coverslips were washed three more times with PEM and permeabilized with 0.1% Triton X-100 for 10 min. After three additional PEM washes, coverslips were blocked in 2 ml PEMBALG (*Woessner and Dawson, 2012*) for 30 min and incubated overnight at 4°C with anti-TAT1 (1:250) and/or anti-GFP (1:500, Sigma) antibodies. The following day, coverslips were washed three times in PEMBALG and then incubated with Alexa Fluor 555 goat anti-rabbit IgG (1:1000; Life Technologies), Alex Fluor 594 goat anti-mouse antibodies (1:250; Life Technologies) and/or Alex Fluor 647 goat anti-mouse (1:250; Life Technologies) antibodies for 2 hr at room temperature. Coverslips were washed three times each with PEMBALG and PEM and mounted in Prolong Gold antifade reagent with DAPI (Life Technologies).

### Flagellar pair length measurements

*Giardia* trophozoites were fixed and stained with TAT1 (1:250) and Alexa Fluor 594 goat anti-mouse IgG (1:250; Life Technologies) as described above. For flagellar pair length measurements, serial sections of immunostained trophozoites were acquired at 0.2 µm intervals using a Leica DMI 6000 widefield inverted fluorescence microscope with a PlanApo × 100, 1.40 numerical aperture (NA) oil-immersion objective. DIC images were analyzed in FIJI (*Schindelin et al., 2012*) using a spline-fit line to trace the flagella from the cell body to the flagellar tip. Flagellar length measurements were analyzed and quantified using custom Python scripts. n ≥ 35 flagella for each pair. Flagellar length data are shown as mean relative length changes with 95% confidence intervals.

### Live imaging of IFT in *Giardia*

For live imaging, strains were grown to confluency, incubated on ice for 15 min to detach cells and pelleted at 900 x g for five minutes at 4°C. Cell pellets were washed three times in 6 mL of cold 1x HBS. After the final wash, cells were resuspended in 1 mL of cold 1x HBS. For live imaging, 500 µL of washed trophozoites were added to the center of a prewarmed 35 mm imaging dish (MatTek Corporation) and incubated for 20 min at 37°C. The imaging dish was washed with three times with warmed 1x HBS to remove unattached cells. For some experiments, CellMask Deep Red plasma membrane stain (ThermoFisher) was used to label the cell membrane (1x final concentration, 15 min at 37°C). Attached cells were embedded in 1 mL 3% low melt agarose (USB Corporation) in 1x HBS (37°C) to limit flagellar beating and prevent detachment. The imaging dish was sealed using parafilm and imaged on a 3i spinning disc confocal microscope (Intelligent Imaging Innovations, Inc).

### Quantification of IFT81mNG full axoneme fluorescence intensity

For analysis of IFT along the entire length of axonemes (basal body to flagellar tip), the IFT81mNG strain was grown to confluency and prepared for live imaging (see above). Images were acquired on a 3i spinning disc confocal microscope as described below. The segmented line tool in FIJI was used to measure the fluorescence intensity along the entire axoneme from basal body to the flagellar tip. Quantification was conducted for the anterior and posteriolateral flagella from 31 cells acquired in three independent experiments. The overall distance from the basal body to the flagellar pore and the flagellar tip was also recorded for each cell. Python scripts were used to plot the mean intensity for all analyzed cells and to calculate and plot the 95% confidence interval for all measurements.

### Fluorescence recovery after photobleaching (FRAP)

To quantify IFT protein of kinesin-13 dynamics at different regions of axonemes, the integrated IFT81mNeonGreen strain or kinesin-13mNG strain was grown to confluency and prepared for live imaging as described above. Images were acquired using a 3i spinning disc confocal microscope (Intelligent Imaging Innovations, Inc) using a 63x, 1.3 NA objective. The microscope was warmed to 37°C one hour prior to image acquisition to maintain cells at physiological temperature and DefiniteFocus (Zeiss) was used to prevent drift during image acquisition. For the IFT81mNG strain, either the cytoplasmic axoneme region or the flagellar pore region of the posteriolateral flagellum was bleached using Vector (Intelligent Imaging Innovations, Inc) with a 488 nm laser (10% laser power, 5 ms exposure). For the kinesin-13mNG strain, the distal tip region of flagella was bleached using Vector (Intelligent Imaging Innovations, Inc) with a 488 nm laser (10% laser power, 1 ms exposure). To monitor recovery, images were acquired with a 488 nm laser (50 ms exposure and 50% laser power, gain set to two with intensification set to 667) at one-second intervals. Images were processed using the Template Matching plugin for FIJI to correct drift during acquisition. Once corrected for drift, ROIs were identified, and intensity measurements were recorded for each experimental time point. Background intensity measurements were taken for each cell analyzed from an adjacent area with no detectable fluorescence. Photobleaching measurements were taken from a non-bleached region of the cell analyzed. Quantification of fluorescence recovery was performed using custom Python scripts to subtract background intensity and correct for photo-bleaching. Individual recoveries were fit using *Equation 1* (*Ellenberg et al., 1997*) to measure the apparent diffusion constant:

$$I_{CA}(t) = I_0 \left( 1 - \sqrt{\frac{\omega^2}{\omega^2 - 4\pi \mathrm{D}t}} \right) \qquad (1)$$

where $I_{CA}(t)$ is the intensity as a function of time and zero time is the bleaching event; $I_0$ is the final intensity after recovery; $\omega$ is the bleach strip width; $\mathrm{D}$ is the diffusion constant.

At short times, *Equation 1* becomes *Equation 2*:

$$I_{CA}(t) = I_0 \frac{2\pi \mathrm{D}t}{\omega_{CA}^2} \qquad (2)$$

where $\omega_{CA}$ is the bleach strip width for the cytoplasmic axoneme. To predict the initial recovery of the flagellar pore region, the flux from retrograde IFT transport is added to *Equation 2*:

$$I_{FP}(t) = I_0 \left( \frac{2\pi \mathrm{D}t}{\omega_{CA}^2} + \frac{\lambda_0}{C_0} \frac{v}{\omega_{FP}} \right) t \qquad (3)$$

where $I_{FP}(t)$ is the intensity as a function of time; $\frac{\lambda_0}{C_0}$ is the relative difference in integrated fluorescence intensity in the membrane-bound axoneme and the cytoplasmic axoneme; $v$ is the speed of retrograde IFT; and $\omega_{FP}$ is the width of the bleach strip for the flagellar pore. Inputting all measured parameters into *Equations 2 and 3* predicts that $\frac{I_{FP}}{I_{CA}} = 3 \pm 1$ for the initial period of recovery. Fits of the initial linear phase of recovery were conducted using linear regression.

All recovery measurements (32 cytoplasmic axonemes and 25 flagellar pores, from at least three independent experiments) were average and rescaled based on the average initial intensity and bleach depth.

## IFT particle tracking using kymograph analysis

To track IFT train traffic in live cells, the integrated IFT81mNeonGreen strain was first grown to confluency and then prepared for live imaging as described above. Images were acquired using a 3i spinning disc confocal microscope (Intelligent Imaging Innovations, Inc) using a 100x, 1.46 NA objective (330 images, 30 ms exposure, intensification = 667, and gain = 2; the total time was about 26 s for each acquisition for a frame rate of approximately 13fps). The microscope was warmed to 37°C one hour prior to image acquisition to maintain cells at physiological temperature and DefiniteFocus (Zeiss) was used to prevent drift during image acquisition.

Kymographs were generated using the KymographClear 2.0 plugin for FIJI (*Mangeol et al., 2016*). A maximum intensity projection image was generated from the time lapse series to identify flagella, and a segmented, spline-fit line was used to trace identified flagella in the time-lapse images. Kymographs were generated for cells that had at least two identified flagella from different flagellar pairs to make intracellular comparisons. Intracellular comparisons were made for 22 cells for the anterior and caudal flagella and for 42 cells for the anterior and posteriolateral flagella, obtained from five independent microscopy experiments. Kymographs were analyzed using KymographDirect software with background correction and correction for photobleaching, and data were further analyzed using custom Python scripts.

## IFT injection frequency distributions

Forward-filtered kymographs generated above were used to measure the time-lag between IFT train injections. Using custom Python scripts, we calculated the time between each injection event for anterior, posteriolateral, and caudal flagella. Frequency histograms for each flagellar pair were fit with a single exponential to measure the rate of IFT injection for each flagellar pair.

## Quantification of integrated IFT particle intensity using super-resolution microscopy

The integrated IFT81GFP strain was first grown to confluency, then fixed and stained as described. 3D stacks were collected at 0.125 µm intervals on a Nikon N-SIM Structured Illumination Super-resolution Microscope with a 100x, 1.49 NA objective, 100 EX V-R diffraction grating, and an Andor iXon3 DU-897E EMCCD. Images were reconstructed in the 'Reconstruct Slice' mode and were only

used if the reconstruction score was 8. Raw and reconstructed image quality were further assessed using SIMcheck and only images with adequate scores were used for analysis (*Ball et al., 2015*).

To determine intensity profiles along the length of flagellar pairs, we used the maximum intensity projections of reconstructed SIM images for tubulin (anti-TAT) and IFT81GFP (anti-GFP). Intensity measurements from ten different cells from three separate experiments were used. Intensity profiles and flagellar length measurements were measured using FIJI and the total integrated intensity was calculated by determining the total area under the curve (AUC) using custom Python scripts.

## Quantification of kinesin13mNG membrane-bound axoneme fluorescence intensity

Trophozoites expressing kinesin13mNG were prepared for live imaging as above. The segmented, spline-fit line tool in FIJI was used to trace the length of the flagellum, from the tip to the base, and measure the intensity. Measurements from at least 23 cells for each flagellar pair from two independent experiments were used. Only cells with at least two measured flagella were analyzed. Custom Python scripts were used to generate plots and statistical analyses of the data.

## Mathematics of kinesin-13 transport

We assume that kinesin-13 on the flagella undergoes directional anterograde motion due to IFT and diffusional motion when it is detached from the IFT train. Furthermore, we assume that freely diffusing kinesin-13 can bind to the anterograde IFT trains anywhere in the flagellum, and that it detaches from this protein complex when it falls apart upon reaching the distal tip. With these simple chemical assumptions we can write an equation for the concentration of kinesin-13 at position $x$ (in µm) away from the flagellar pore as:

$$\frac{dc(x)}{dt} = D\frac{d^2c(x)}{dx^2} - k_{on}c(x). \tag{4}$$

The first term represents diffusion of kinesin-13 along the flagellum, while the second term represents binding to the IFT train. Each binding event removes a kinesin-13 molecule from the diffusing pool, which is then transported to the flagellar tip by anterograde IFT, where it is released.

A steady state concentration of kinesin-13 is reached on the time scale set by the transport of kinesin-13 to the flagellar tip, which is a few seconds. In steady state, the solution to *Equation 4* is $c(x) = c_0 \cosh\left(\frac{x}{\lambda}\right)$ where $c_0$ is the concentration of kinesin-13 at the flagellar pore, and $\lambda = \sqrt{\frac{D}{k_{on}}}$ is the decay length. If we make the simple assumption that the rate at which kinesin-13 is captured by anterograde IFT trains is set by the time required for a kinesin-13 molecules to diffuse to the IFT particles, which are localized to the microtubules within the flagellum, we obtain $k_{on} = D/R^2$. Here $R$ is the radius of the flagellum, which is on the order of a few hundred nanometers, and it is the typical distance traveled by a diffusing kinesin-13 before it is captured by IFT. Putting this estimate for $k_{on}$ into the equation for the decay length, we obtain $\lambda = R$.

Our simple model of kinesin-13 transport, which only assumes diffusion of kinesin-13 in the flagellum, and binding to the anterograde IFT complex, which carries it to the flagellar tip before the kinesin-13 is released there, makes two key quantitative predictions: (*i*) kinesin-13 is localized to the tip of the flagella and its concentration profile decays exponentially away from the tip. (*ii*) The decay length of the kinesin-13 gradient is given by the size of the flagellar cross-section radius and therefore it is independent on the length of the flagellum. We find both predictions to be borne out by our data (*Figure 8E and F*). This quantitative agreement between theory and experiment provides indirect evidence for our transport model of kinesin-13.

So far, we have implicitly assumed an infinite supply of kinesin-13 by assuming that its concentration at the flagellar pore ($c_0$) stays constant as the flagellum grows. If we incorporate the constraint that the total amount of kinesin-13 in the flagellar pore plus the amount in the flagellum stays constant during ciliogenesis, we arrive at the formula:

$$c(L) = c_{\text{init}} \frac{e^{\frac{L}{\lambda}}}{1 + v \, e^{\frac{L}{\lambda}}} \tag{5}$$

where $c_{\text{init}}$ is the initial concentration when all the kinesin-13 is in the flagellar pore, and $v$ is the ratio

of the volume of a stretch of flagellum of length $\lambda$, and the volume of the cellular reservoir from which kinesin-13 is taken up during ciliogenesis (either the cytoplasm or the flagellar pore). For flagella that are a few microns in length, the quantity $\frac{1}{v}$ is the enhancement factor, which describes how much the concentration at the tip of the flagellum is greater than its initial concentration at the flagellar pore (i.e., the effect of the combined anterograde transport and diffusion concentrates kinesin-13 at the tip). Given that the decay length is a few hundred nanometers, as is the radius of the flagellum, and assuming that the reservoir of kinesin-13 is a few microns in diameter, we estimate $v \approx 10^{-3}$. We use these order of magnitude estimates to plot the theoretically predicted concentration profile of kinesin-13 in *Figure 7A* and the predicted disassembly rate $k_- \, c(L)$ (see next section, *Equation 6*) in *Figure 7B*.

## Mathematics of flagellar length control by length-dependent disassembly

We describe the growth of a flagellum as a competition between assembly and disassembly. Assembly is IFT dependent. If the IFT flux into the flagellum of length $L$ is $I$, then, assuming first-order chemical kinetics for the loading of the tubulin to the IFT, the rate of assembly is $k_+ = I \, (N - L)$. $N$ is the total amount of tubulin available for assembly, so $N - L$ represents the amount of free tubulin.

We further assume that the disassembly of microtubules occurs at the distal tip of the flagellum and is mediated by kinesin-13 binding at the tip. Assuming first order kinetics for this binding process, the rate of disassembly is proportional to the concentration of kinesin-13 at the tip: $k_- \, c(L)$; $c(L)$ is given in Equation 6 and $k_-$ is a second order rate constant (with units per M per second).

Putting assembly and disassembly into a single equation for the flagellar length gives

$$\frac{dL}{dt} = I \, (N - L) - k_- \, c(L), \tag{6}$$

which is graphically represented in *Figure 7B*. The equation simply states that the rate of change of flagellar length is given by the difference between the rate of assembly and the rate of disassembly.

Steady state is reached when assembly and disassembly are balanced, which occurs at length $L^* = N - k_- c(L^*)/I$. From this equation we conclude that longer flagella have smaller concentration of kinesin-13 at the tip, as observed (*Figure 6D,E*).

### Data availability

All code and data are available via GitHub: https://github.com/shanemc11/Giardia_Flagella (copy archived at https://github.com/elifesciences-publications/Giardia_Flagella).

## Acknowledgements

This work was supported by NIH/NIAID awards 2R01AI077571-10A1 to SCD. JK is funded by the National Science Foundation grants DMR-1610737 and MRSEC-1420382, and by the Simons Foundation. SGM was supported by NIH T32 GM0007377. Plasmid pKS_mNeonGreen-N11_NEO was a gift from Alex Paredez (University of Washington, Seattle). We thank the MCB Light Microscopy Imaging Facility, a UC Davis Campus Core Research Facility, for the use of the 3i spinning disc confocal microscope (Intelligent Imaging Innovations, Inc) and the N-SIM Structured Illumination Super-resolution Microscope (Nikon). We thank the Physiology course at the Marine Biological Laboratory for insightful discussions and Kari Hagen for valuable editorial assistance.

## Additional information

### Funding

| Funder | Grant reference number | Author |
| --- | --- | --- |
| National Institutes of Health | 2R01AI077571-10A1 | Scott C Dawson |
| National Institutes of Health | T32 GM0007377 | Shane G McInally |
| National Science Foundation | DMR-1610737 | Jane Kondev |

| National Science Foundation | MRSEC-1420382 | Jane Kondev |
| Simons Foundation | | Jane Kondev |

The funders had no role in study design, data collection and interpretation, or the decision to submit the work for publication.

## Author contributions
Shane G McInally, Conceptualization, Resources, Data curation, Formal analysis, Investigation, Visualization, Methodology; Jane Kondev, Conceptualization, Formal analysis, Visualization, Methodology; Scott C Dawson, Conceptualization, Supervision, Funding acquisition, Visualization, Project administration

## Author ORCIDs
Shane G McInally (iD) https://orcid.org/0000-0001-6145-4581
Jane Kondev (iD) http://orcid.org/0000-0001-7522-7144
Scott C Dawson (iD) https://orcid.org/0000-0002-0843-1759

## Decision letter and Author response
Decision letter https://doi.org/10.7554/eLife.48694.sa1
Author response https://doi.org/10.7554/eLife.48694.sa2

# Additional files
## Supplementary files
• Transparent reporting form

## Data availability
All data generated or analysed during this study are included in the manuscript and supporting files.

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
