## [Decision Letter]

**Acceptance summary:**

The lengths of eukaryotic cilia are tightly controlled by balancing assembly and disassembly. In *Chlamydomonas*, intraflagellar transport-mediated assembly appears to be the critical parameter that controls length. Work presented here indicates that Giardia uses an alternative method for length control. This parasite has four pairs of cilia that are templated from basal bodies in the cell. Unlike cilia in *Chlamydomonas* or mammals, Giardia cilia have a long stretch of axoneme extending through the cytoplasm before the cilia project from the flagellar pore and become typical membrane covered organelles. Each of the four pairs of cilia has a distinct length that is distributed between the cytoplasmic and membrane-enclosed compartments. The authors demonstrate that intraflagellar transport rates and frequency are similar between cilia of different lengths but the level of kinesin-13 is inversely correlated with length. Since kinesin-13 can depolymerize microtubules, the authors propose a model where length dependent disassembly regulates the length of each pair of cilia. Future studies will be needed to understand how the cell can deliver specific amounts of kinesin-13 to each of the flagellar pairs.

**Decision letter after peer review:**

Thank you for submitting your article "Length-dependent disassembly maintains four different flagellar lengths in Giardia" for consideration by *eLife*. Your article has been reviewed by three peer reviewers, and the evaluation has been overseen by a guest Reviewing Editor and Anna Akhmanova as the Senior Editor. The reviewers have opted to remain anonymous.

The reviewers have discussed the reviews with one another and the Reviewing Editor has drafted this decision to help you prepare a revised submission.

Summary:

This interesting manuscript from McInally and colleagues examines the regulation of flagellar length in Giardia. Giardia has four pairs of flagella each with a distinct steady-state length. This raises the question of how a cell is able to regulate the length of its individual cilia. In *Chlamydomonas*, ciliary length is regulated by intraflagellar transport but the authors conclude that injection rates are similar in all cilia suggesting that this is not the mechanism in Giardia. The authors then focus on the role of microtubule depolymerization in length regulation. They find that loss of the depolymerizing kinesin-13 increased the length of all flagella and that kinesin-13 accumulates at the tips of flagella in an inverse relationship to flagellar length. From this work, a model is developed whereby ciliary length is regulated by the kinesin-13 driven disassembly at the ciliary tip. This work highlights the evolutionary adaptations to ciliary assembly and disassembly that haven been utilized by different organisms.

Essential revisions:

1) Data supporting the model that ciliary tip kinesin-13 levels regulate ciliary disassembly to control ciliary length need to be strengthened. While the data shows that the amount of kinesin-13 at the tip is inversely correlated with ciliary length, remaining parts of the model are largely unsupported. As the model depends upon IFT delivery of kinesin-13 to the ciliary tip and diffusion back, these parameters need to be demonstrated. In addition, levels of tip kinesin-13 during de novo cilia assembly needs to quantitated, as this is critical to the model. Measuring the dynamics of tubulin at the tips of the different cilia by photo bleach of fluorescent tubulin (see Hao et al., NCB2011 for example) may also provide support for the model.

2) The manuscript needs significant editing for brevity and clarity. Each of the reviewers had specific instances where they had difficulty. In addition, the limited precursor and unlimited precursor models are not well described and it is not clear which model is supported by the authors. Data showing that kinesin-13 is inversely related to ciliary length seems to support the limited precursor model but the failure to see changes upon taxol treatment suggests the unlimited precursor model. Figure 8C suggests that ciliary tip kinesin-13 is the same in long and short cilia contradicting the data in the paper and the statement "Both models assume that a higher disassembly rate is conferred by more kinesin-13 localized to shorter flagella (caudal, posteriolateral) than to longer (ventral, anterior) flagella during de novo assembly."

[Editors' note: further revisions were requested prior to acceptance, as described below.]

Thank you for submitting your article "Length-dependent disassembly maintains four different flagellar lengths in Giardia" for consideration by *eLife*. Your revision has been reviewed by the original peer reviewers, and the evaluation has been overseen by a guest Reviewing Editor and Anna Akhmanova as the Senior Editor.

The reviewers have discussed the reviews with one another and the Reviewing Editor has drafted this decision to help you prepare a revised submission.

The reviewers and I appreciate the effort that you and your colleagues took to revise this manuscript. We all feel that it is substantially improved. Unfortunately, our main concern regarding the strength of the data supporting your model remains. That said we believe the work is important and provides new ideas for the field. While we cannot accept the manuscript as it is currently written, if you can revise the work to either strengthen support for your model or tone down the conclusions so they are better aligned with the data we will be happy to reconsider this decision.

In particular, the statement from the Abstract "Rather than by length-dependent IFT-mediated assembly, we demonstrate that Giardia maintains four different flagellar lengths by modulating length dependent kinesin-13-mediated disassembly." should be modified to summarize the findings from the kinesin-13 studies and be followed by a more subdued conclusion. In addition, the Discussion should be expanded to make it clear that there are alternatives to kinesin-13 in regulating ciliary length.

---

## [Author Response]

Essential revisions:1) Data supporting the model that ciliary tip kinesin-13 levels regulate ciliary disassembly to control ciliary length need to be strengthened. While the data shows that the amount of kinesin-13 at the tip is inversely correlated with ciliary length, remaining parts of the model are largely unsupported. As the model depends upon IFT delivery of kinesin-13 to the ciliary tip and diffusion back, these parameters need to be demonstrated.

While IFT transport of kinesin-13 has not been directly demonstrated in Giardia, previous studies in *Chlamydomonas* (Piao et al., 2009) have demonstrated that kinesin-13 utilizes IFT for transport to the flagellar tip. However the manuscript has been revised to emphasize the direct tests of specific predictions from our model of flagellar length control. Specifically, we experimentally demonstrate the following predictions from our model:

1) The fluorescence intensity of kinesin-13 is greater at the distal tips of shorter flagella (Figure 6E).

2) The distribution of kinesin-13 within the flagellum is exponential (Figure 8E).

3) The exponential decay is length-independent (Figure 8F).

4) Kinesin-13 undergoes turnover at the distal tip with a small mobile fraction (Figure 8G, 8H, Figure 8—figure supplement 2).

5) Taxol elongation of flagella does not change the fluorescence intensity of kinesin-13 at the distal tip (Figure 8D, Figure 8—figure supplement 1A).

6) Prolonged exposure to Taxol increases the length of all flagella by similar amounts (Figure 8B).

*In addition, levels of tip kinesin-13 during* de novo *cilia assembly needs to quantitated, as this is critical to the model. Measuring the dynamics of tubulin at the tips of the different cilia by photo bleach of fluorescent tubulin (see Hao* et al., *NCB2011 for example) may also provide support for the model.*

We agree that quantification of kinesin-13 at the flagellar tips during ciliogenesis would be informative, however it is technically unfeasible. de novo flagellar assembly occurs immediately prior cytokinesis in Giardia. Giardia cultures cannot be synchronized, and trophozoites divide rapidly. To date, only a single study has successfully imaged dividing Giardia during and immediately after cytokinesis over four hours (Hardin et al., 2017). Furthermore, only two of the four flagellar pairs are assembled de novo following cytokinesis (the ventral and posteriolateral, Nohynkova, Euk Cell, 2006); the others are structurally inherited. It is likely that ablation methods will be necessary to assess de novo ciliogenesis of all flagellar pairs, however these techniques have not yet been developed for Giardia, and our initial attempts at ablation were unsuccessful likely due to phototoxicity of imaging anaerobic protists following ablation. We anticipate considerably more troubleshooting to develop this technique in Giardia.

We also agree that photobleaching of tubulin at the tips would be informative. Yet photobleaching tubulin at flagellar tips in tubulin-GFP strains has also proven problematic, possibly due to phototoxic effects during the monitoring of long term recovery (>20 minutes) and other issues regarding live imaging in Giardia (ex: complete immobilization of flagella). We anticipate further methodologic development for tubulin-specific FRAP and for live imaging kinesin-13 localization during cell division and have revised the Discussion accordingly.

2) The manuscript needs significant editing for brevity and clarity. Each of the reviewers had specific instances where they had difficulty.

We have substantially edited the manuscript for brevity and clarity.

*In addition, the limited precursor and unlimited precursor models are not well described and it is not clear which model is supported by the authors. Data showing that kinesin-13 is inversely related to ciliary length seems to support the limited precursor model but the failure to see changes upon taxol treatment suggests the unlimited precursor model. Figure 8C suggests that ciliary tip kinesin-13 is the same in long and short cilia contradicting the data in the paper and the statement "Both models assume that a higher disassembly rate is conferred by more kinesin-13 localized to shorter flagella (caudal, posteriolateral) than to longer (ventral, anterior) flagella during* de novo *assembly."*

We have substantially edited the manuscript for brevity and clarity.

[Editors' note: further revisions were requested prior to acceptance, as described below.]

The reviewers and I appreciate the effort that you and your colleagues took to revise this manuscript. We all feel that it is substantially improved. Unfortunately, our main concern regarding the strength of the data supporting your model remains. That said we believe the work is important and provides new ideas for the field. While we cannot accept the manuscript as it is currently written, if you can revise the work to either strengthen support for your model or tone down the conclusions so they are better aligned with the data we will be happy to reconsider this decision.In particular, the statement from the Abstract "Rather than by length-dependent IFT-mediated assembly, we demonstrate that Giardia maintains four different flagellar lengths by modulating length dependent kinesin-13-mediated disassembly." should be modified to summarize the findings from the kinesin-13 studies and be followed by a more subdued conclusion. In addition, the Discussion should be expanded to make it clear that there are alternatives to kinesin-13 in regulating ciliary length.

We again appreciate the enthusiasm and feedback on this manuscript. In this resubmission we have:

1) Revised the statement in the Abstract “"Rather than by length-dependent IFT-mediated assembly, we demonstrate that Giardia maintains four different flagellar lengths by modulating length dependent kinesin-13-mediated disassembly." to reflect that we find the correlation observed that kinesin-13 localization to the flagellar tips is inversely correlated to flagellar length.

2) Revised and expanded the Discussion to further include alternative to kinesin-13 to regulating ciliary length. Due to limitations in experimental methodologies for live imaging in Giardia, we have chosen to “tone down” conclusions rather than develop additional methods and provide additional data. These revised discussion points and alternatives models are primary summarized and focused in the last section of the Discussion: “Control of flagellar length by a limiting precursor pool”.